# Understanding the diurnal cycle of land-atmosphere interactions from flux site observations

Eunkyo Seo[1,2], Paul A. Dirmeyer[1]

[1] Center for Ocean-Land-Atmosphere Studies, George Mason University, Fairfax, 22030, United States

[2] Department of Environmental Atmospheric Sciences, Pukyong National University, Busan, 48513, Republic of Korea

*Correspondence to*: Eunkyo Seo (eseo8@gmu.edu)

**Abstract.** Land–atmosphere interactions have been investigated at daily or longer time scales due to limited data availability and large errors for measuring high-frequency variations. Yet coupling at the sub-daily time scale is characterized by the diurnal cycle of incoming solar radiation and surface fluxes. Based on flux tower observations, this study investigates the climatology of observed land–atmosphere interactions on sub-daily time scales during the warm season. Process-based multivariate metrics are employed to quantitatively measure segmented coupling processes and mixing diagrams are adopted to demonstrate the integrative moist and thermal energy budget evolution in the atmospheric mixed layer. The land, atmosphere, and combined couplings for the entire daily mean, midday, and midnight show different situations to which surface latent and sensible heat fluxes are relevant, and they also reveal the climate sensitivity to soil moisture and surface air temperature. The 24-hour coevolution of the moist and thermal energy within the boundary layer traces a particular path on mixing diagrams, exhibiting different degrees of asymmetry (time-shifts) in water– and energy–limited locations. Water– and energy–limited processes also show opposing long tails of low humidity during the daytime and night-time, related to the impact on land and atmospheric couplings of latent heat flux and other diabatic processes like radiative cooling. This study illustrates the necessity of considering the entire diurnal cycle to understand land-atmosphere coupling processes comprehensively in observations and models.

## 1 Introduction

Land–atmosphere (L–A) interactions play a critical role in the global energy and water cycles. Our understanding of L–A interactions has increased greatly over the last 20 years, initially via numerous climate modelling studies. These have included several multi-model experiments (Koster et al., 2011, 2002; Dirmeyer et al., 2006; Seneviratne et al., 2013; Lawrence et al., 2016; Van Den Hurk et al., 2016; Xue et al., 2016, 2021), and single model studies too numerous to mention. Among the most important multi-model studies was the Global Land–Atmosphere Coupling Experiment (GLACE), which focused on how land surface states (namely soil moisture) can affect atmospheric processes (Koster et al., 2004, 2006; Guo et al., 2006), leading to the identification of hotspot locations of L–A coupling.

In recent years, the growing availability of observational data (both *in situ* measurements and satellite retrievals) has made possible a new wave of research that is enhancing our understanding of L–A interactions and enabling more thorough evaluations of model performance. Growing *in situ* monitoring networks of soil moisture are enabling new evaluation capabilities (Dorigo et al., 2011; Quiring et al., 2016; Dirmeyer et al., 2016). Flux towers have reached a level of quality, coverage and longevity that make them invaluable to studies of L–A interactions (Novick et al., 2018; Tramontana et al., 2016). Satellites are providing ever improving coverage and quality of land surface states, and increasingly fluxes (Miralles et al., 2016; Alemohammad et al., 2017; Colliander et al., 2017; Dorigo et al., 2017; Ma et al., 2019; Seo and Dirmeyer, 2022). Tawfik et al. (2015b) demonstrated linkages between land surface fluxes and convective initiation from radiosonde data. Denissen et al. (2021) found soil moisture signals globally in boundary layer profiles. Zhang et al. (2020) have applied sounding data from commercial aircraft to quantify land surface drivers of boundary layer development. Dirmeyer et al. (2018) verified L–A coupling in forecast models, reanalyses, and land surface models against *in situ* observations using process-based multivariate statistics, demonstrating that the models generally underrepresent spatial and temporal variability relative to observations. Wulfmeyer et al. (2018) are developing a new generation of surface and lower atmosphere monitoring capabilities that will provide unprecedented data on local L–A interactions. Data assimilation and other synthesis techniques can extend the data coverage while compensating for both model and sensor errors (Crow et al., 2015; Reichle et al., 2017; Seo et al., 2021).

Moreover, the increased data availability of hydrological and near-surface atmospheric variables can be used to improve understanding of L–A interactions following links in the process chains described by Santanello et al. (2018) . The linkages begin with soil moisture and its controls on surface heat flux partitioning, its effects on soil heat storage, conduction, and the health of vegetation. This process chain proceeds through near surface atmospheric states, boundary layer properties, cloud formation and convection. These strongly influence L–A feedbacks in the development of extreme climate events such as heat waves and drought (Seneviratne et al., 2010; Miralles et al., 2012, 2019; Schumacher et al., 2019; Seo et al., 2020; Dirmeyer et al., 2021). These couplings are not necessarily linear, and the soil moisture–evaporation relationship is found to strengthen when the soil moisture and temperature become drier and warmer, respectively, which emphasizes anomalous warming and drying to the extreme (Benson and Dirmeyer, 2021). Thus, a realistic representation of L–A coupling in a subseasonal-to-seasonal forecast system is key to improved prediction skill (Seo et al., 2019; Koster et al., 2011).

Most L–A coupling metrics (refer to Table 1 in Santanello et al. (2018)) have focused on daily mean conditions, using data that was commonly available from models when L-A interaction studies began. However, some metrics use information at specific times of day to focus on time-evolving processes within the diurnal cycle. For instance, the mixing diagram, an integrative diagnostic metric of the L–A coupling process chain, demonstrates the daytime coevolution of energy and water budgets within the mixed layer (ML) (Santanello et al., 2009, 2011). This synthesized metric can be decomposed into land and atmospheric components that are further explained by linked moist and thermal processes to quantify interactions and feedbacks across a range of scales. The convective triggering potential (CTP) and low-level humidity index ($HI_{low}$) characterize the circumstances in which the L–A coupling could influence afternoon convection (Findell and Eltahir, 2003b, a). They are

based on the concept that morning atmospheric profiles of temperature and humidity can provide information on whether ML conditions are favourable to trigger convection during the day. Findell et al. (2011) established that increased morning evaporation leads to an enhanced probability of afternoon rainfall for the boreal summer season over much of the United States, whereas rainfall intensity appears insensitive to surface fluxes. The heated condensation framework (HCF) also examines the impact of surface fluxes on convective triggering later in the day based on a synthetic evolution of atmospheric profiles of temperature and humidity driven by idealized surface fluxes (Tawfik and Dirmeyer, 2014; Tawfik et al., 2015a). The climatological probability of summertime convective initiation was found to be more sensitive to morning convective inhibition over the southeastern United States, while soil moisture provides a secondary control on convection (Tawfik et al., 2015b). In addition, the influence of soil moisture on cloud development has been demonstrated for the coupled L–A system with realistic daytime surface fluxes and atmospheric profiles (Ek and Holtslag, 2004) and the role of dry-air entrainment has been shown to enhance surface evaporation and induce a shallower convective boundary layer through daytime L–A feedbacks (Van Heerwaarden et al., 2009).

Nevertheless, thorough examinations of the climatology of the complete diurnal cycle of L–A interactions have been lacking. A major barrier has been the availability of reliable data that resolves the diurnal cycle, particularly for near-surface soil moisture. Although dielectric sensors have been extensively used in soil moisture monitoring networks for the past few decades, their diurnal cycle at shallow soil depths includes a spurious component due to temperature sensitivity, causing a positive measurement bias that peaks during the time of maximum soil temperature (Kapilaratne and Lu, 2017). To date, there is no adequate temperature correction method for dielectric sensors (M. Cosh, personal communication), so typically hourly or sub-hourly measurements are averaged to daily means, or measurements at a single hour of the day are used, avoiding the sensor bias problem. Although cosmic ray neutron sensors do not have this problem (Zreda et al., 2008; Evans et al., 2016), those sensors have a variable measurement footprint and depth, and are not as widely used due to their expense. Polar orbiting satellites with sun-synchronous overpasses near sunrise and sunset provide data at the same hour of the day (Entekhabi et al., 2010; Kerr et al., 2010). However, they do not sample the entire diurnal cycle, at best providing measurements twice per day at any location, and depending on latitude, may only pass over a location every few days. There are also inconsistencies between morning and evening overpasses (Leroux et al., 2013).

By considering the issues included in diverse observational datasets, the investigation of the complete diurnal cycle of L–A interactions can begin to provide a comprehensive understanding of the L–A coupling processes. In this study, we examine the entire diurnal cycle of climatological L–A interactions at available flux tower sites across the globe in a way that ameliorates the problems described above. The terrestrial coupling index is adopted in an hour-by-hour context to explore the L–A coupling process chain. The mixing diagram approach is extended around the full diurnal cycle to synthesize the coevolution of moist and thermal energy budgets within the ML. We sidestep the soil moisture bias problem by grouping data by each hour of the day and calculate correlation-based daily coupling metrics independently at each hour. In so doing, new details of the daily evolution of L–A coupling are revealed. Section 2 introduces the datasets used in this study. Section 3 describes the adopted metrics to understand the L–A interactions, and our composition approach to investigate the climate

sensitivity. Section 4 presents and discusses the results of this study. Finally, section 5 summarizes the results and their implications for future applications.

## 2 Data

### 2.1 Flux site observations

In situ measurements of near-surface meteorological variables, land surface heat fluxes, and surface soil moisture are employed to understand L–A interactions on sub-diurnal time scales. The FLUXNET2015 station dataset version released in February 2020 has collected data from multiple regional flux networks across the globe spanning 1996–2020 (https://fluxnet.org/data/fluxnet2015-dataset/; (Pastorello et al., 2020). The tier 1 data is used in this study, additionally screened by the quality flags for each variable marked 0 or 1 (0: measured and 1: good quality gap-filled value following the method of Reichstein et al. (2005)). In addition, if the IGBP classification of any sites is snow and ice (IGBP classification is "SNO"), the sites are discarded. To extend the observational flux data across more stations and into more recent years, this study also uses data from the AmeriFlux network (https://ameriflux.lbl.gov/) and the European Drought-2018 network (https://doi.org/10.18160/YVR0-4898). Data from these additional sources are available in a format that matches the FLUXNET2015 standards.

To examine the diurnal cycle of L–A interactions, this study uses half-hourly or hourly data from all three network datasets and composites all sites to hourly intervals. Where FLUXNET2015 spatially and temporally overlaps the AmeriFlux or European Fluxes Database station data, the FLUXNET2015 is given priority and the other datasets are used to extend the temporal coverage of the FLUXNET2015 data. Fig. 1 shows the global distribution of sites along with their land cover categories. 230 sites are available, but the spatial coverage is concentrated in midlatitude regions, especially over North America, Europe, and Australia. The adopted variables in this study are soil wetness content in the top soil layer (SWC1), sensible (H) and latent heat fluxes (LE), surface air temperature, humidity, surface pressure, and vapor pressure. Except for SWC1, the other variables are assumed to have been measured a few meters above the canopy while acknowledging that the canopy height varies among sites. However, the flux observations generally do not contain the canopy information necessary to compare to the reference height of the sensors, which is a shortcoming of using flux tower data, especially for forested locations. To understand the atmospheric coupling processes related to land surface heat fluxes, we calculate the lifted condensation level (LCL) using the near-surface measurements at each site. The LCL can be characterized as a potential level of cloud base formation based on parcel theory, and is easily calculated from surface meteorological measurements, but is an approximation subject to the limitations of parcel theory. In reality, the profile of temperature and moisture above the surface also determine the level of the cloud base (Tawfik and Dirmeyer, 2014). The LCL can be compared to the planetary boundary layer (PBL) height to define an LCL deficit (PBL height minus LCL; Santanello et al., 2011). When the PBL grows to the height of the LCL (corresponding to positive values of the LCL deficit), water may condense from the air parcel, and cloud

formation occurs, although clouds begin to form when scattered updrafts penetrate the condensation level before the entire ML
reaches the LCL (Van Stratum et al., 2014). The LCL is formulated as:

$$LCL = \frac{T - T_d}{\Gamma_d - \Gamma_{dew}} \tag{1}$$

where $T$ and $T_d$ are surface air temperature and dew-point temperature, respectively. The terms of $\Gamma_d$ and $\Gamma_{dew}$ are the lapse rate for dry adiabatic lifting (9.8×10$^{-3}$ K/m), and the lapse rate of the dew point (1.8×10$^{-3}$ K/m), respectively. LCL is reported in units of meters.

As the instrument height varies among flux towers, this study computes the measurement height ($h$) as the difference between reported height of observation and averaged canopy height. When 83 observation sites (36% of the total) do not provide both heights, we assume the measurement height as the averaged value across the other available sites (7.5 m). All flux measurements are taken above the canopy while few meteorological sensors are below the canopy top. Many flux observation sites in forests report 2-m meteorological quantities that are not consistent with the WMO standard for unobstructed routine measurements upon which the mixing diagrams were originally developed. To understand the possible effect of sub-canopy measurements on the diurnal mixing diagrams, meteorological data from the Discovery Tree at the Andrews Experimental Forest (https://portal.edirepository.org/nis/mapbrowse?packageid=knb-lter-and.5476.2; Site and Still, 2019) have been used. The Discovery Tree is a 50-m tree in the Willamette National Forest in the western US that has been instrumented with numerous sensors. Data from 1 October 2015 through 5 December 2018 at 1.5 m and 56 m above the ground (below and above the canopy) are employed in this study. Although temperature and humidity variables are available, air pressures at both layers are adopted as constant values because this site does not measure pressure at each vertical level – this has minimal impact on the results.

## 2.2 ERA5 reanalysis

Information on PBL height ($Z_{PBL}$) is needed in the mixing diagram approach described in section 3.3, in order to estimate the temperature and humidity budgets in the ML. However, flux tower sites do not typically measure PBL height. This study alternatively adopts $Z_{PBL}$ from the European Centre for Medium-Range Weather Forecasts (ECMWF) Reanalysis version 5 (ERA5; Hersbach et al., 2020) on the model's native grid, corresponding to a horizontal spatial resolution of ~25 km and an hourly temporal resolution. $Z_{PBL}$ from the ERA5 grid cell containing each flux site location is associated with that location. Although there are some issues in downscaling the gridded data to the observed sites due to unresolved spatial heterogeneity in the atmospheric boundary layer, Vilà-Guerau De Arellano et al. (2020) found a satisfactory agreement between ERA5 and three independent observations, which demonstrates that the boundary layer shows similar temporal evolution on the larger regional scale. Additionally, the inter-comparison of daytime $Z_{PBL}$ from four reanalysis datasets against globally distributed high-resolution radiosonde measurements suggests that the most accurate reanalysis product is ERA5 (Guo et al., 2021).

## 3 Methodology

### 3.1 Data pre-processing

Coupling metrics are calculated separately for each month to remove the seasonal cycle, and then monthly statistics are averaged for each hemisphere's warm season (NH: May–September, SH: November–March) to focus on the most active season for L–A coupling. However, it should be noted that the temporal data coverage for each flux site varies greatly; some stations have more than two decades of data, others only a few years. Moreover, to avoid the confounding effects of precipitation on correlation-based metrics, substantial rainfall days are identified when daily soil moisture tendencies are positive and larger than 2-standard deviations; those days are removed from the calculations. Only when all 24-hourly values are available for a given day are they included in the analysis.

### 3.2 Terrestrial coupling index

To quantify L–A interactions, this study uses the terrestrial coupling index, proposed by Dirmeyer (2011), to characterize the sensitivity of the target variable (i.e., land surface fluxes) to the representative variability of the source variable (i.e., soil moisture). It is formulated as:

$$TCI_h(SV_h, TV_h) = R(SV_h, TV_h) \times SD(TV_h) \qquad (2)$$

where $SV$ and $TV$ are the source and target variables, respectively, and the subscript $h$ refers to the local hour of the day. The terms $R$ and $SD$ are the temporal correlation coefficient, and the temporal standard deviation of the corresponding time series, respectively. $TCI$ is calculated using day-to-day time series grouped by local hour $h$, so that 24 separate coupling indices are calculated at each flux site for each month. This approach avoids the aforementioned problem of spurious diurnal soil moisture biases due to the dielectric sensor errors; the daytime bias is ameliorated by only combining data from the same time each day, and correlations are insensitive to the absolute magnitudes of data, thus minimizing the contribution of diurnal sensor errors. Depending on the source and target variables, we can define different land and atmospheric coupling indices. For the land leg, SWC1 is commonly the source variable, and either H or LE is the target variable. These two land couplings are referred to as $L(SWC1, H)$ and $L(SWC1, LE)$. For the atmospheric leg, LCL is chosen as the target variable and H and LE are the source variables; the two atmospheric couplings are $A(H, LCL)$ and $A(LE, LCL)$. Additionally, Dirmeyer et al. (2014, see also Lorenz et al. 2015) extended the terrestrial coupling index and proposed the integrative L–A feedback metrics by combining the land and atmospheric legs. This quantifies the two-legged coupling process initiated from soil moisture variability, carried through to the response of the atmosphere. It is formulated as:

$$TCI_h(SV_h, IV_h, TV_h) = R(SV_h, IV_h) \times R(IV_h, TV_h) \times SD(TV_h) \qquad (3)$$

where $IV$ is the intermediate variable, here the surface fluxes. The two-legged coupling process is mediated by LE or H and source and target variables are always SWC1 and LCL, respectively. They are referred to as the total couplings $T(SWC1, H, LCL)$ and $T(SWC1, LE, LCL)$, the first indicating a pathway via the energy cycle, and the second through the water cycle.

As the sensitivity of the land, atmospheric, and two-legged couplings is not symmetric depending on the pathway through different land surface fluxes, this study investigates their asymmetric behaviour in different coupling segments on the sub-daily time scale.

### 3.3 Mixing diagrams

A mixing diagram is a diagnostic thermodynamic relationship among components of the local L–A coupling process used to understand the integrative moist and thermal energy budget evolution in the ML. It was first introduced by Stommel (1947), who addressed the coevolution of 2-m potential temperature ($\theta$) and humidity ($q$) to the energy and water budgets during daytime PBL growth as a trajectory in a two-dimensional phase space of heat and water. Mixing diagrams break down the evolution of $\theta$ and $q$ into land and atmospheric components in which the flux contributions of surface heat (sensible) and
moisture (latent) result in a land vector in the phase space whose slope corresponds to the Bowen ratio. The remaining components of their trajectories result from various atmospheric process (relevant to PBL entrainment, advection, condensation, evaporation and radiative transfer) (Betts, 1992). Modelling studies have shown the sensitivity of the coevolution of $\theta$ and $q$ to land and boundary layer physics schemes can be evaluated directly against observations (Santanello et al., 2009, 2011).

As near-surface or ML temperature and humidity, surface fluxes, and PBL height information are required to construct a mixing diagram, this integrative metric can also be applied with other data sources such as in-situ flux observations and ground-based active remote sensing products. Therefore, this study employs flux site observations to depict the observed coevolution of $\theta$ and $q$ within the PBL on sub-diurnal time scales. Flux sites provide surface air temperature (which is converted to $\theta$) and $q$, atmospheric pressure, and when $q$ is not available, vapor pressure deficit (VPD which is used along with pressure and temperature to calculate vapor pressure, and then $q$). $\theta$ and $q$ are converted to energy variables, via multiplication by the specific
heat capacity of air ($C_p$=1005 J/kg·K) and the latent heat of vaporization ($L_v$ =2.5x10$^6$ J/kg) respectively. A mixing diagram is constructed with hourly vectors ($V(t)$, $t$ is the local hour), which consist of changes in thermal (specific dry enthalpy) and moisture (water vapor latent heat content) terms on the y- and x-axes respectively: $[\theta(t + 1) - \theta(t)]C_p$ and $[q(t + 1) - q(t)]L_v$. These terms are broken down into the hourly land and atmospheric vector components in this thermal-
moisture phase space.

    For the estimation of the land surface contributions to PBL heat and humidity in the mixing diagram methodology, the vertically averaged temperature and pressure are needed within the PBL to estimate the mean PBL air density ($\bar{\rho}$). These are not available from near-surface measurements at flux towers. The temperature at the PBL top ($T_{PBL}$) is approximated by applying a temperature lapse rate of 6.5 K/km at the $Z_{PBL}$ and the ML temperature ($\bar{T}$) is defined by the average of the surface
air temperature ($T_{air}$) and the PBL temperature. The vertical pressure gradient ($dP/dZ = -\rho g$ where $P$, $Z$, and $g$ are air pressure, vertical depth, and gravitational acceleration of 9.8 m/s$^2$, respectively) and the ideal gas law ($P = \rho RT$ where is gas

constant of 287.058 J/kg·K) are used to obtain the pressure at the PBL top. When the density term in the vertical pressure gradient equation is replaced by the ideal gas law, we obtain:

$$\frac{dP}{dZ} = -\frac{Pg}{RT} \tag{4}$$

Taking the integral of both sides, the pressure at the PBL can be estimated as:

$$P_{PBL} = P_{sfc} e^{-\frac{g(Z_{PBL}-h)}{R(T_{PBL}-T_{air})} \ln\frac{T_{PBL}}{T_{air}}} \tag{5}$$

The mean ML pressure ($\bar{P}$) is approximated by the average of atmospheric pressure ($P_{sfc}$) and $P_{PBL}$. Then, the hourly ML air density ($\bar{\rho} = \bar{P}/R\bar{T}$) is recovered using the ideal gas law. Based on these estimated variables, the hourly land vector component (units: J/kg/hr) consists of surface heat ($F_{sfc}$) and moisture ($M_{sfc}$) terms attributed to sensible and latent flux contributions to the PBL. They are formulated following Santanello (2009):

$$F_{sfc}(t) = \frac{\bar{H}(t)}{\bar{\rho}(t) \, Z_{PBL}(t)} \Delta t \tag{6}$$

$$M_{sfc}(t) = \frac{\overline{LE}(t)}{\bar{\rho}(t) \, Z_{PBL}(t)} \Delta t \tag{7}$$

Each is calculated from hourly averaged sensible ($\bar{H}$) and latent ($\overline{LE}$) heat fluxes where $\Delta t$ is one hour, i.e., 3600 seconds.

Next, the hourly atmospheric vector components are calculated as residuals of the hourly total vectors minus the land vectors, also consisting of surface heat ($F_{atm}$) and moisture ($M_{atm}$) terms. Both the thermal term and moisture term are implicitly defined by entrainment at the top of the boundary layer, horizontal advection, and phase changes of water in the ML. The thermal term for the atmosphere also includes the effects of radiative heating, cooling, and frictional warming. Their formulations are followed as:

$$F_{atm}(t) = C_p[\theta(t+1) - \theta(t)] - F_{sfc}(t) \tag{8}$$

$$M_{atm}(t) = L_v[q(t+1) - q(t)] - M_{sfc}(t) \tag{9}$$

Furthermore, the timely accumulated heat ($\sum F$) and moisture ($\sum M$) terms for the land and atmospheric component, respectively, are defined to characterize the accumulated diurnal budgets in the ML. They are formulated as:

$$\sum F_{comp}(t) = \sum_{h=0}^{t} F_{comp}(h) \tag{10}$$

$$\sum M_{comp}(t) = \sum_{h=0}^{t} M_{comp}(h) \tag{11}$$

where $comp$ is either $sfc$ or $atm$ and $h$ is the hour, accumulations begin at 0000 LST.

One thing that should be remembered is that the '2-m assumption' for $\theta$ and $q$ is embedded in this approach. The original concept for the mixing diagram is that $\theta$ and $q$ represent mean values within the ML. Using near-surface values to represent mean ML values assumes a perfectly mixed ML, introducing some error into the calculations. For instance, surface air temperature is higher (lower) during daytime (night-time) than that in the ML. The large near-surface radiative cooling at night is significant even though this is quite decoupled from the ML. Thus, the '2-m assumption' leads to amplified (reduced) budgets in the mixing diagram during daytime (night-time) for the atmospheric vectors, whereas the land vectors are not affected by this assumption as they are defined by the surface fluxes. Overall, the adaptation of the flux site data is an alternative approach to understand the observed climatology of the coevolution of moist and thermal energy budgets in the ML because there are difficulties to estimate the $Z_{PBL}$ and to observe the vertical $\theta$ and $q$ profiles. However, this does not prevent exploration of the general characteristics of the diurnal cycle and the precise comparison of a model to observations is still possible if one also uses the near-surface variables from the model.

## 3.4 Methodology to separate water– and energy–limited regimes

This study attempts to understand the local sensitivity of the L–A coupling processes in different climate regimes using the analysis approach described above, as the effects of mesoscale meteorology are difficult to isolate. Water- and energy-limited regimes, which indicate whether land heat fluxes are sensitive to the variability of the soil moisture, are categorized at the observed sites to investigate the climate sensitivity of L–A interactions. The proxy to separate the regimes is the temporal correlation between daily mean time series of SWC1 and evaporative fraction $EF = LE/(H + LE)$, which bridges heat and moisture fluxes. Large positive correlations indicate a strong dependence of EF on variations in SWC1, signifying a water-limited regime; negative correlations suggest an energy-limited regime (Dirmeyer et al., 2000; Dong et al., 2022). This study compares the sensitivity of L–A interactions to those different regimes between the top and bottom 10% of the observation sites sorted by the value of this correlation. When the correlation is higher than 0.36 and the corresponding $p$-value is less than 0.005 (also requiring a sufficient sample size at the flux site), the sorted observations are representative of the water-limited regime. When the correlation is lower than 0.08 and the corresponding $p$-value is lower than 0.14, the sorted observations are defined as representative of the energy-limited regime.

## 4 Results

### 4.1 Asymmetric coupling behaviour at sub-daily time scales

To illustrate the diurnal variability of L–A coupling processes, we provide a comparison of the coupling metrics for the daily mean, midday, and midnight periods for the different land surface fluxes (i.e., LE and H). Soil moisture has a proportional relationship to LE based on the water balance, which results in positive values of the coupling metric $L(SWC1, LE)$ when

energy is not limited. As increasing LE leads to a decrease of H via the energy balance, $L(SWC1, H)$ is typically negative. Most of the flux sites show these physical tendencies (Fig. 2a), which are related to the fact that many of the sites are located in summertime water-limited regimes that correspond to "hot spots" of L–A coupling (Dirmeyer, 2011). The land coupling term shows a statistically significant negative relationship between $L(SWC1, H)$ and $L(SWC1, LE)$ for the daily mean, midday, and midnight periods. However, the characteristics of $L(SWC1, H)$ and $L(SWC1, LE)$ are not simply symmetric to each other. For instance, although they have in common that midday coupling variability is greater than that of the daily mean or midnight due to large net radiation, $L(SWC1, LE)$ shows little mean difference across all periods and nested distributions across sites, whereas mean $L(SWC1, H)$ shows larger differences and clear shifts in distributions (Fig 2a). This means that the asymmetry of $L(SWC1, H)$ in the sub-daily time scale is larger than that of $L(SWC1, LE)$, which is mainly attributed to the diurnal reversal of H (positive during the day and negative at night). This characteristic is explored in more detail later.

Fig. 2b shows the atmospheric couplings. The relationship between $A(H, LCL)$ and $A(LE, LCL)$ is not significant during midday, based on a high p-value along with low correlation, due to their opposite relationships on either side of $A(LE, LCL) = 0$. This is clearly shown in their density functions, in which $A(LE, LCL)$ has peaks on both positive and negative sides of zero even though $A(H, LCL)$ has only one peak on the positive side. The positive $A(LE, LCL)$ and $A(H, LCL)$ situation occurs in energy-limited locations whereby increased net radiation leads to increasing LE and H along with rising temperature, which subsequently induces an LCL increase. In contrast, the result in the negative $A(LE, LCL)$ case is explained by the water-limited processes such that decreasing LE leads to decreasing relative humidity and dew point temperature, which subsequently induces an LCL increase. These physical atmospheric coupling processes are not seen during the night-time; the daytime processes dominate the daily mean results. Moreover, if $A(H, LCL)$ is greater than $A(LE, LCL)$, it means that the boundary layer is more sensitive to H than LE, and vice versa when $A(LE, LCL) > A(H, LCL)$. Higher $A(H, LCL)$ during the daytime is due to the stronger correlation between H and LCL, and higher $A(LE, LCL)$ during the night-time is attributed to the negative values of H.

The observed two-legged couplings from soil moisture to LCL, mediated by H and LE, are mostly on the left side of y=-x line (Fig. 2c). Most couplings are negative, which means LCL height is anticorrelated with soil moisture regardless of the pathway of the coupling. To the left of the y=-x line (octants IV through VII), points in octants VI and VII indicate stronger two-legged coupling of soil moisture on potential cloud base through H rather than through LE. Locations presenting stronger coupling through H are almost two times more common than through LE throughout the entire day. This arises mainly from the larger correlation in the terms of land and atmosphere coupling via H; the LCL is less sensitive to LE variability in dry land conditions (not shown). Although there is a clear difference in both two-legged couplings between midday and midnight, the density distributions of the LE-mediated coupling for the daily mean and the midday are similar. In contrast, those related to H are quite different: the midday result exhibits more negative mean coupling and is more widely distributed. Both are commonly negatively skewed across the entire sub-daily time span, and they are attributable mainly to the atmospheric leg of coupling.

The land coupling tends to be stronger when the climate is relatively warm and dry, and the effect is more pronounced during midday than midnight (Figs. 3a and 3d). Although there is a clear difference between $A(LE, LCL)$ and $A(H, LCL)$ for midday and midnight, the climate sensitivity of both atmospheric couplings according to the range of soil moisture is very different for moisture versus energy coupling pathways (Fig. 3b). For instance, the response of $A(LE, LCL)$ to changing soil moisture shows negative values as the soil dries due to water limitations, and positive values that increase as the soil gets wetter due to energy limitations (c.f., Fig. 2b). However, $A(H, LCL)$ is much less sensitive.

In contrast, the sensitivity of $A(H, LCL)$ to temperature is comparable to $A(LE, LCL)$ and the moisture pathway results from the soil moisture state (warm temperatures usually correspond to dry soil). The midday $A(H, LCL)$ coupling strength decreases as temperatures warm, but the coupling is dramatically increased in the warmest category, in which the $A(LE, LCL)$ becomes negative (Fig. 3e). The H-driven coupling sensitivity is attributed to the temperature sensitivity within the correlation $R(H, LCL)$. The incoming radiation in warm climates is mostly transferred to H, limiting any humidity increase, which results in strong sensitivity between H and LCL. The daily mean reflects the midday result as there is little contribution from overnight processes.

Meanwhile, the sensitivity of the two-legged couplings to soil moisture and temperature also differs, and their characteristics are most pronounced in the daytime. The sensitivity of $T(SWC1, LE, LCL)$ to soil moisture during midday (Fig. 3c) is high in relatively dry climates despite less change in $T(SWC1, H, LCL)$, which is mostly attributed to the effects of the atmospheric leg. Conversely, in relatively wet climates, $T(SWC1, H, LCL)$ is highly sensitive to soil moisture despite muted changes in $T(SWC1, LE, LCL)$, as $L(SWC1, H)$ represents a larger contribution to the sensitivity to soil moisture than does $L(SWC1, LE)$. The midday results have a similar sensitivity to the daily mean despite the lack of sensitivity at night. The results categorized by temperature show strong coupling in $T(SWC1, H, LCL)$ and $T(SWC1, LE, LCL)$ only for the warmest days during both midday and midnight because of the temperature sensitivity in the atmospheric coupling (Fig. 3f). There are also categorical differences in coupling sensitivities across different land covers (Fig. S1). For instance, wetlands generally agree with the results for wet and cold climates and coupling for savanna sites is consistent with the results shown in Fig. 3 for dry and warm climates.

## 4.2 Diurnal mixing diagrams

In this subsection, we explore the full 24-hour cycle of mixing diagrams for a comprehensive understanding of the water and energy budget evolution in the boundary layer relevant to the L–A interactions. First, the diurnal L–A coupling terms are averaged across 230 observation sites to illustrate climatological behaviour (Fig.4). Panels a-c are constructed in the same manner as mixing diagrams, with moisture variability along the x-axis and heat variability on the y-axis, but instead plot the daily evolution of the two-legged, terrestrial, and atmospheric couplings, respectively. During the daytime, both two-legged couplings are negative, with $T(SWC1, H, LCL)$ being almost three times as strong as $T(SWC1, LE, LCL)$ around midday,

showing the importance of sensible heating for ML growth (Fig. 4a). It is consistent with the result that three times more locations exhibit stronger two-legged coupling of soil moisture to LCL through H than through LE (c.f., Fig. 2c). The sign of

the two-legged coupling is determined by the multiplication of the correlation terms, representing land and atmospheric couplings. $R(SWC1, H)$ and $R(H, LCL)$ are mostly distributed on negative and positive sides during the daytime, respectively, leading to consistently large magnitudes. On the contrary, $R(SWC1, LE)$ and $R(LE, LCL)$ span 3 of the 4 quadrants, so do not result a consistent sign, reducing the mean magnitude of the moist pathway (Fig. S2). Both two-legged metrics contain the same standard deviation term, so it is the correlations that lead to larger negative mean values of $R(SWC1, H) \times R(H, LCL)$

than the corresponding pathway via LE. There is an asymmetry in the path of moisture and temperature across the diurnal cycle, in that the extremes in the thermal process chain lead the moist process chain by 2-3 hours. As a result, the evening path through the water-energy phase space does not retrace the morning path. The thermal coupling collapses toward zero quickly in the late afternoon, while the moist coupling declines gradually throughout the evening.

In the land leg, $L(SWC1, H)$ and $L(SWC1, LE)$ attain large negative and positive values respectively during the daytime, with

the stronger $L(SWC1, H)$ about double the magnitude of $L(SWC1, LE)$ (Fig. 4b). The diurnal growth and decay of the coupling strengths also exhibit some asymmetry with the phase of $L(SWC1, H)$ leading $L(SWC1, LE)$ by about an hour, in contrast to the surface fluxes themselves wherein the thermal fluxes lead the moisture fluxes. This results from the asymmetry of $R(SWC1, LE)$ between developing (morning) and decaying (afternoon) phases, whereas $R(SWC1, H)$ is relatively symmetric (not shown). H and LE peak in the early and late afternoon, respectively, each strongly controlled by gradients between the

land surface and lower atmosphere. As the air warms in the afternoon and incoming solar radiation starts to decline, the thermal gradient weakens reducing H. At the same time, the warm air increases the potential evaporation by maintaining a large vapor pressure deficit, facilitating strong rates of LE. Couplings peak about noon and are near zero throughout the night-time hours. For the atmospheric couplings (Fig. 4c), there is a more complex evolution. $A(H, LCL)$ is positive during the day and negative at night, while $A(LE, LCL)$ is positive across the entire day. Each reaches a maximum during the early afternoon, and the

coupling strength of $A(H, LCL)$ is double that of $A(LE, LCL)$, due to higher correlation $R(H, LCL)$. The diurnal coevolution reveals asymmetry with abrupt decaying of $A(H, LCL)$ from 3-7 PM. $A(LE, LCL)$ peaks in strength about 4 PM dropping quickly to a minimum at 8 PM before beginning a gradual 20-hour rise. The diurnal atmospheric coupling asymmetry is determined by the evolution of $R(H, LCL)$ and $R(LE, LCL)$, which reveals the phase of $A(H, LCL)$ leading $A(LE, LCL)$ by 2-3 hours, and emphasized by the large daytime LCL variability. It is characterized by the diurnal maximum of LCL variance at

3-4 PM and its abrupt decaying from that maximum. The result is a figure-eight path in Fig.4c.

The observationally-based diurnal mixing diagram (Fig. 4d) shows the climatological coevolution of moisture and thermal energy budgets within the ML. The path of the ML specific dry enthalpy and water vapor latent heat content trace a banana-shaped pattern, with a strong diurnal cycle of heat content, but a clear semi-diurnal cycle for moisture, driven by the interplay of changing surface evaporation and depth of the ML. Note that the daily means are not enveloped within the hourly path on

the mixing diagram. The daily mean ML potential temperature and humidity are not experienced at the same time at any hour of the day, exposing the problem of using daily mean data to assess L-A coupling. Furthermore, the ML budget exchange

processes experience strong asymmetry. There is commonly an increase in both moist and thermal energy per unit mass from 4 AM to 8 AM, after which moist energy decreases until 3 PM while thermal energy continues increasing. This is followed by a decrease in thermal energy while the moisture energy increases until 7 PM and then decreases until the next morning.

To identify the distinct roles of land and atmosphere in the diurnal mixing diagram evolution, we examine the hourly component vectors from surface fluxes (Fig. 4e) and atmospheric processes (Fig. 4f). On average across every hour of the day, moisture is supplied by surface evaporation (Fig. 4e). Daytime evaporation and transpiration are strong, but appear to continue during night-time in the hourly mean data. Meanwhile, there is thermal energy loss during the night-time and gain during the daytime. The net changes during the entire day attributed to land surface processes can be defined by the vector from the origin

to point 23. The length of each hourly vector indicates the rate of change of heat content contributed by surface fluxes, portrayed in Fig. 4g in terms of energy per unit mass of air per hour. The rates are highest in the morning and gradually decrease in the afternoon because $Z_{PBL}$ reaches a peak around 2 PM, maximizing the volume of the reservoir accepting the surface fluxes. Because of the strong relationship between the mean and variance of land heat fluxes, the corresponding land couplings also have a strong correlation with the mean values. However, the land vectors are somewhat different from the land couplings

since the vectors are also affected by the diurnal variability of $Z_{PBL}$ (Eq. 6 and 7). The time series of diurnal land tendencies in Fig. 4g outline an ellipse, $M_{sfc}$ remaining positive at all hours. Moisture and thermal tendencies abruptly increase at sunrise, reaching a maximum in early to mid-morning with $M_{sfc}$ peaking about two hours before $F_{sfc}$, then both tendencies gradually decrease until midnight.

On the other hand, the accumulated atmospheric components (Fig. 4f) show a gradual daylong decrease in moist energy, while

there is gradual thermal energy increase from sunrise to midday, then a decrease until midnight. The only moistening through the day is very small, occurring between 5-7 AM and around 5 PM (Fig. 4h). The positive temperature and negative moisture tendencies from 7 AM to the noon are mostly related to entrainment of drier air with higher potential temperature at the top of the growing boundary layer. The negative tendency of thermal energy from the afternoon onward is likely dominated by radiative cooling (Betts et al., 1996), although advection, entrainment, and phase changes due to condensation, precipitation

and reevaporation may also contribute. Drying in the afternoon is likely due to net moisture diffusion into the free atmosphere from the ML, and removal of water vapor from the air by condensation in clouds. These effects combine to produce an omega-shaped path in the diurnal atmospheric components (Fig. 4f). Although the daily mean is not enveloped within the hourly evolution on the mixing diagram (Fig. 4d), the daily mean values of both land and atmospheric vector components are enveloped by their diurnal paths (Fig. 4g and h), emphasizing that the ML budget exchange processes at sub-daily time scales

is a complex interaction of surface and atmospheric processes.

**4.3 Climate regime dependence**

Additionally, we examine the sensitivity of the diurnal budget coevolution and the L–A interactions separately for water– and energy–limited regimes. Based on the aforementioned approach to separate these regimes (Sec. 3.4), we have composited the upper and lower 10% of the sites. The average soil moisture and temperature of the water–limited observation sites are 0.13 m$^3$/m$^3$ and 23.6 °C, respectively; for energy–limited sites they are 0.29 m$^3$/m$^3$ and 19 °C, respectively.

The three segments of diurnal L–A couplings over the water–limited regions show different sub-daily pathways and stronger couplings than for the energy–limited sites (Fig. 5). Although the coupling strengths for both sets are maximum during the daytime, the diurnal coevolution of two-legged couplings (Fig. 5a) in the water–limited sites resembles more closely the climatological series (cf., Fig. 4a), but stronger, while the energy–limited sites have very weak couplings. For the land couplings, the diurnal behaviour for the water–limited sites shows characteristically negative $L(SWC1, H)$ and positive $L(SWC1, LE)$ with comparable coupling strengths between them (Fig. 5b). Over energy–limited sites, SWC1 and LE are anticorrelated as evaporation controls soil moisture. Dry soils still correspond to deeper boundary layers, but the magnitudes of the coupling metrics are a fraction of their moisture-limited counterparts. The closer to the water–limited regime, the higher the magnitude of the correlation between SWC1 and both surface fluxes. Neither of these extreme composites shows much diurnal asymmetry.

For the climate sensitivity of atmospheric couplings (Fig. 5c), there is a strong divergence of behaviours. $A(H, LCL)$ over the water–limited regime is stronger than over the energy–limited regime, which results from the larger LCL variability along with the marginal sensitivity of $R(H, LCL)$ to the climate regime. Both show a diurnal evolution of $A(H, LCL)$ that is negative at night, grows strongly positive through the morning peaking a couple hours after local noon. However, $A(LE, LCL)$ is highly divergent between the water– and energy–limited regimes. While comparable in magnitude, water limited regimes show anticorrelation between surface evaporation and LCL height throughout the day, peaking twice (around noon, then more strongly at sunset), while the energy-limited regime registers positive correlations all day and a single mid-afternoon peak. The water–limited result for $A(LE, LCL)$ is attributed to the proportional relationship of LE as a source of water vapor to relative humidity and dew point temperature, leading to an anticorrelation with LCL height (c.f., Fig. 2b). The results for energy-limited sites are not attributable to direct surface forcing of LCL or ML characteristics, but rather the dominance of atmospheric dynamics and circulation in determining both near-surface meteorology and surface flux rates. Warm periods correspond to more net radiation and stronger evaporation at the same time the LCL is higher, while cool moist periods limit both LCL height and latent heat flux.

The observed diurnal mixing diagrams also exhibit banana-shaped paths, but the water– and energy–limited regions reveal different long tails (Fig. 5d). Both show a morning peak in ML humidity, but the driest time for the ML is during early afternoon in moisture-limited regimes and before sunrise in moisture-limited regimes. Both regimes span mostly the same range of water vapor latent heat content, but they have little overlap in terms of dry enthalpy. Also, the daily means are not enveloped within the hourly path for either regime, but lie closer to their respective paths compared to Fig. 4d. Dry regimes also exhibit much

greater asymmetry. The differences are mostly induced by differing moist budget evolution in land and atmospheric components. For instance, the daytime long tail is related to the small moisture increase due to the relatively smaller mean LE accompanying soil dryness, so that atmospheric entrainment induces strong drying in the ML. The early morning long tail in energy-limited regimes appears dominant due to the large moisture budget decrease via atmospheric effects during the afternoon (Fig. 5h), while there is a large moisture increase by the land surface along with a reduced moisture decrease by the

atmosphere from 4 AM to 8 AM (Fig. 5g,h).

To identify the distinct climate sensitivity of land and atmosphere in the mixing diagrams, we examine the hourly component vectors (Fig. 5e and f). Despite the comparable incoming net radiation at the land surface, the partitioning of the net radiation to LE and H (i.e., the Bowen ratio) differs between the climate regimes, which results in a respective net gain and loss in the heat and moisture budgets across the entire day. In the water–limited regime, the arid surface conditions lead to less LE, with

the extra energy going toward H, which drives the large increases in thermal energy during the daytime even though there is a larger loss of thermal energy during the night-time (larger negative H; Figs. 5e and 5g). For energy-limited regimes, moisture fluxes are larger and thermal fluxes are smaller.

The atmospheric components for the diurnal mixing diagram (Fig 5f and h) also show a distinct climate sensitivity in the ML moisture dimension even though the climate regimes are separated by the characteristics of land coupling processes described

above. In the energy–limited regime, the positive moisture tendency due to evaporation around sunrise starts earlier than for the moisture-limited regime since the sun rises earlier on average at the energy-limited flux sites as they tend to be at higher latitudes (Fig. 5h). Moreover, there is a larger negative moisture tendency from afternoon to the next early morning, which characterizes the larger atmospheric moisture loss over the energy–limited regime (Fig. 5f). Interestingly, the moisture limited regimes show two periods of atmospheric-driven ML moistening during the day: from 5-7 AM but also from 4-9 PM. We can

only speculate on the causes in the composites, but investigation of individual flux tower sites in semi-arid regions near evaporative moisture sources (e.g., irrigated farmland) show evidence of moist advection during the afternoon (not shown).

## 4.4 Canopy effects

Canopy dynamics are important in forest areas. For instance, the upper canopy is covered by most of the leaf area that absorbs

most of the solar radiation, which explains the great majority of carbon and water cycling over the forest. Thus, the observations of canopy microclimate can advance a biological understanding of canopy processes and their interactions to atmosphere in terms of sub-canopy dynamics (Site and Still, 2019; Senécal et al., 2018). The variation of diurnal mixing diagrams as a function of the position in the canopy of the instrumentation is investigated here for a single site in a midlatitude forest region. The diurnal mixing diagrams from below and above the canopy both exhibit banana-shaped paths (Fig. 6). However, unlike

the climatological result averaged across globally distributed sites, at this location there is found a clockwise diurnal trace due to a large diurnal cycle of latent heat flux that peaks more strongly in the evening than the morning. The timing of the diurnal extremes of both temperature and moisture are the same at both levels except for the humidity maxima, which are each two

hours later above the canopy than near the surface, presumably driven by the extra evaporation of canopy interception (dew) in the morning and transpiration into a stable boundary layer around sunset. Otherwise, the region above the canopy is both

warmer and moister throughout the daylight hours. Entrainment affects the air above the canopy before the sheltered air near the surface, so that the peak moisture content of the air above the canopy occurs around 8 AM, but at 10 AM near the surface. Moreover, the daytime exhibits a large thermal energy discrepancy between the sunlit canopy and shaded ground along with comparable moisture content, which accounts for a higher relative humidity below the canopy.

On the other hand, the night-time shows a strong moisture content difference along with relatively smaller thermal energy

contrast than the daytime. The cooling and large moisture decrease from sunset to the next morning is attributed to the atmospheric effects (c.f., Fig. 4f) as radiative cooling leads to saturation and dew formation. The air above the canopy at night remains warmer than the air near the ground at this location, perhaps due to the sizeable heat capacity of the biomass of this old growth forest and its ability to retain daytime heat in the upper canopy (Swenson et al., 2019). Nevertheless, the overall progressions of the diurnal cycle above and below the canopy share the same main features in the mixing diagram, despite

their relative displacements. These results are for a single site – locations with the necessary data to calculate mixing diagrams at multiple vertical levels in a canopy appear to be quite scarce.

## 5 Summary

Most previous studies exploring L–A interactions have been restricted to daily or lower frequency time domains because of

the limited availability of data resolving the diurnal cycle and inherent sensor issues that make it difficult to measure sub-daily variability. Although coupling characteristics between the daytime and night-time are obviously different due to the large disparity in available energy, namely incoming solar radiation, this research area has been underexplored. Now, there are an increasing number of long-term flux tower datasets available measuring land surface and near-surface atmospheric variables at hourly or finer time resolution across the globe. The baseline for such observational datasets is FLUXNET2015; the

AmeriFlux and the European Fluxes Database are additionally used to extend data availability spatially or temporally for this study. Here, we have described the climatology of the observed L–A interactions at sub-daily time scales during the local summer season across 230 sites (Fig. 1).

To measure the response of the target variables to the representative variability of the source variables in the L-A coupling paradigm (Santanello et al., 2018) in a chain from land states to surface fluxes and atmospheric characteristics, this study

adopts multivariate metrics that define land, atmospheric, and combined couplings through both the water and energy cycles. To understand the heat and moisture budget exchanges within the ML, the mixing diagram approach has been adapted to extend the relationship between the coevolution of the budgets and L-A couplings across the entire day. We have quantified the mean conditions across sites and distributions, with a particular focus on the most water-limited and energy-limited locations with regard to surface fluxes. We find the diurnal cycles of both mixing diagrams and hourly L-A couplings usually

exhibit asymmetry between the water and energy cycles. Using hourly observations, information from the coupling metrics

and mixing diagrams has been synthesized to reveal in great depth the evolution of L-A interactions across the diurnal cycle, and to differentiate unique behaviours in energy-limited and water-limited regimes.

Segmented coupling metrics for the land leg ($L$), the atmospheric leg ($A$), and joint two-legged ($T$) metrics are compared among entire daily mean, daytime, and night-time periods for moisture (LE) and thermal (H) pathways. The land leg couplings (Fig. 2a) show significant negative relationships between $L(SWC1, LE)$ and $L(SWC1, H)$ across sites for the daily, midday, and midnight averages. This result is explained by the proportional relationship between soil moisture and LE based on the water balance equation and the negative relationship between soil moisture and $H$. The diurnal land coupling evolution exhibits an asymmetry with the phase of $L(SWC1, H)$ leading $L(SWC1, LE)$ by an hour or two. $L(SWC1, H)$ mostly attains negative values regardless of the hour of day and background climate whereas $L(SWC1, LE)$ is negative and positive in the energy-limited and water-limited regimes, respectively. The land couplings tend to be stronger where the climate is warmer and drier (water-limited regimes), also evident in Figs. 3a, 3d and 5b, and the effect is most pronounced during the daytime.

Regarding atmospheric couplings, the diurnal phase shift in $A(H, LCL)$, which shows daytime positive and night-time negative correlations, is consistent regardless of the climate regimes (Fig. 5c). The coherent night-time negative correlation is attributed to the physical process chain such as large negative H, indicating a large temperature gradient between colder ground and warmer air (Fig. S3b). $A(H, LCL)$ is rather insensitive to soil moisture variations, and daytime $A(H, LCL)$ tends to weaken as mean temperature increases up to the warmest category ($T > 26˚C$) where coupling strength abruptly increases (Fig. 3e). In contrast, $A(LE, LCL)$ reveals a clear sensitivity to climate regime because as LE decreases, LCL necessarily increases in water-limited locations, but where energy is limited, meteorological variations drive both LCL height and evaporation rates (Fig. 5c). The diurnal atmospheric coupling evolution represents positive and negative peaks around early afternoon and midnight, respectively, and exhibits asymmetry with the phase of $A(H, LCL)$ leading $A(LE, LCL)$ by about two hours. Moreover, the atmospheric couplings (especially the correlation component) commonly weaken nonlinearly whereas the functional relationship of H is stronger than that of LE (Fig. S3).

The corresponding integrated two-legged couplings, $T(SWC1, LE, LCL)$ and $T(SWC1, H, LCL)$, are mostly negative (Fig. 2c), meaning dry soils correspond to a higher cloud base. The stronger daytime values of $T(SWC1, H, LCL)$ suggest variations in H exert more control on LCL than variations in LE. The daytime values of $T(SWC1, LE, LCL)$ and $T(SWC1, H, LCL)$ are highly sensitive to soil moisture variations toward the dry and wet ends of the soil moisture distribution, respectively (Fig. 3c), marking very different behaviours between dry and wet regimes, but there is little sensitivity to the temperature except, again, at the warm extreme (Fig. 3f). The stronger two-legged couplings in a warm and dry climate (water-limited regime) result from the combination of larger negative correlation ($R(SV, IV) \times R(IV, TV)$) and higher variability of the LCL.

In many previous studies, only a daytime budget analysis using mixing diagrams has been conducted, but this study covers the entire diurnal cycle. The results of the full diurnal mixing diagrams (Figs. 4 and 5) show that the path of ML specific dry enthalpy and water vapor latent heat content across all 24 hours traces a banana shaped path, and the different phases of heat (a single peak in early afternoon) and moisture (a double peak) mean the daily average state of the ML is not actually

experienced at any hour of the day (Fig. 4d). The diurnal mixing diagram breaks down the hourly vector of $\theta$ and $q$ into land and atmospheric components.

The land vector components show added moisture from evaporation across the entire day, but a thermal energy gain (loss) during the daytime (night-time) depending on the sign of $H$ (Fig. 4e). Thus, the net contribution of LE to the total daily energy budget in the ML is larger than from H. The individual diurnal evolutions of surface fluxes and PBL depth result in a maximum of positive humidity and temperature tendencies during the morning (Fig. 4g). The peak hourly coupling strength occurs after the maximum heat and moisture tendencies occur.

The diurnal atmospheric components are calculated as residuals of the mixing diagram minus land surface flux contributions, and represent a synthesis of many effects (e.g., PBL entrainment, horizontal advection, radiative cooling, and etc), which produces an "omega" shaped path in hourly atmospheric vectors of ML humidity and temperature (Fig. 4f). The effect of atmospheric entrainment is greatest during the period of ML growth in the morning, when the entrained dry and high potential temperature air at the top of the PBL causes positive temperature and negative moisture tendencies in the ML. Entrainment weakens but continues after the ML reaches maximum depth until dissipation of the daytime boundary layer around sunset. The atmospheric entrainment characterizes the maximum of both tendencies around noon and the stronger negative moisture tendency (Barr and Betts, 1997). However, the impact of the entrainment is mainly from 7 AM to noon. Meanwhile, there is radiative cooling of the ML at all hours that there are no clouds above the ML trapping longwave radiation. The radiative cooling likely dominates afternoon when mean tendencies become negative (Fig. 4h). When the net ML dry enthalpy supplied by entrainment is near its diurnal maximum, the atmospheric couplings tend to be strongest.

The water– and energy–limited processes represent a large discrepancy in the ML specific dry enthalpy despite a small difference of water vapor latent heat content (Fig. 5d). The 24-hour mixing diagram for water-limited processes exhibits much greater asymmetry in the water-energy phase space across the entire day. The climate regimes also exhibit opposing long tails of minimum water vapor content: whether a location experiences the driest ML just before sunrise or in the afternoon depends on the balance of competing drivers: land surface evaporation adding moisture and entrainment mixing dry air into the ML. In water-limited regimes, entrainment dominates and minimum $L_v q$ occurs when dry enthalpy peaks. In energy-limited regimes, minimum $L_v q$ occurs when the air is coolest, consistent with the Clausius–Clapeyron relationship in which the temperature decrease reduces the water-holding capacity of the air. Regarding the climate sensitivity in the land component vectors, the partitioning of the net radiation into LE and H shows correspondence to the climate regimes (Fig. 5e). In a water–limited regime, larger H and smaller LE during the daytime lead to a larger maximum, net, and range of thermal energy than in the energy–limited regime, but greater moistening across the day in the energy-limited regime. The difference in net moisture and thermal energy gain depends on the climate regime: the larger being around 60% greater than the smaller in each regime. Interestingly, despite having smaller net surface radiation during the day, energy-limited regimes appear to have a greater 24-hour net surface energy contribution from the land surface $\sum(M_{sfc} + F_{sfc})$ than moisture-limited regimes due to their higher total evaporation, shallower ML, and less overnight sensible heat transfer from atmosphere to land; the difference in energy per unit mass is around 35-40% (comparing markers labelled "23" in Fig. 5e). Despite the PBL in water–limited regimes being

about twice as deep as for energy-limited regimes, with accompanying stronger entrainment, the impact of the atmospheric entrainment over both climate regimes is similar, resulting in a positive temperature and a negative moisture tendency from 7

AM to the noon.

**6 Conclusions**

Overall, this study suggests there is more to be learned about L–A interactions by the comprehensive study of sub-daily time scales. The asymmetric diurnal evolution of the land, atmosphere, and combined coupling metrics as well as within the 24-

570    hour path of ML water and energy content portrayed in the mixing diagrams begs further study. With additional data, particularly profile measurements within and above the atmospheric boundary layer, it would be possible to begin to decompose the atmospheric evolution into its component terms, separating advection from entrainment and other diabatic processes. We can imagine a role for single-column models as useful diagnostic tools to aid further study. The metrics introduced in this study could also be applied to understand and evaluate the diurnal cycle of L–A interactions in models.

Essentially, this study makes the case for the need to attend to sub-daily processes for a better understanding of L–A coupling, even while much research is still focused on evaluations based on daily data. This study is also of potential value for future atmospheric model development, such as for PBL and convective parameterizations in mesoscale models on a sub-daily time scale. Data availability remains a limitation; we hope work such as this can motivate the collection of more data that resolves the diurnal cycle over land.

Additionally, this study has examined the dependence of the diurnal cycle in mixing diagrams above and below the canopy at a single old-growth forest site. The comparison shows that clear contrasts due to the thermal heat content are strongest during the day, but they exist across the diurnal cycle. Meanwhile, at this one site, the contrasts in moisture content are greatest at night. Currently available FLUXNET data does not allow for such examination of contrasts in the climatological diurnal cycles of heat and water throughout the canopy, nor the separate impacts of transpiration, canopy interception and surface evaporation.

No doubt there are interesting regional differences dependent on climate regimes and biomes that remain unknown. Although the very tall, dense, old growth forest site examined here may represent an extreme case of contrast between near-surface and above-canopy mixing diagrams, the results suggest the need to examine further canopy dynamics for the fundamental understanding of processes and properties across a range of forested regions.

Lastly, it should be noted that the diagnoses presented here presume the fidelity of the flux tower measurements, but there are

known biases and a distinct lack of energy balance closure at most sites (Cheng et al., 2017). The assumptions of Monin-Obukhov similarity theory, widely applied for flux tower calculations and in many model parameterizations, are compromised in many situations (Wulfmeyer et al., 2018) including variations across the diurnal cycle and inconsistencies between moisture and thermal fluxes (Van De Boer et al., 2014). These problems may affect details of the diurnal cycles in the figures presented here, particularly when trends or rates of change are marginal. However, we feel the main features and contrasts shown here

are likely robust, and certainly worth closer investigation. Mean biases do not affect correlations or standard deviations, which

are at the heart of the coupling metrics, but diurnally dependent biases could affect some results presented here. Within the limits already inherent in coupling metrics, the results presented here are consistent with current process understanding yet shed new light on the relationships between energy and water cycles, between land and atmosphere, by combining and extending existing approaches in a novel way.

## Acknowledgements

This study was supported by NOAA grant NA19NES4320002 to the Cooperative Institute for Satellite Earth System Studies (CISESS) at the University of Maryland/ESSIC via a subaward (79785-Z755420) to the Center for Ocean-Land-Atmosphere Studies at George Mason University. We also wish to thank Hsin Hsu for constructive comments which helped us improve the manuscript.

## Code availability

The source code used in this study is shared on the GitHub (https://github.com/ekseo/Diurnal_LA_coupling.git).

## Data availability

Flux tower observations that support the findings of this study are openly available in from the FLUXNET2015 Tier 1 data (https://fluxnet.fluxdata.org/), the AmeriFlux network (https://ameriflux.lbl.gov/), the drought-2018 network (https://doi.org/10.18160/YVR0-4898), and the Discovery Tree data at the Andrews Experimental Forest (https://portal.edirepository.org/nis/mapbrowse?packageid=knb-lter-and.5476.2). The Copernicus Climate Change Service (C3S) provides access to ERA5 data freely through its online portal at https://cds.climate.copernicus.eu/cdsapp#!/dataset/reanalysis-era5-single-levels.

## Author contributions

ES led manuscript writing and performed most of the data analysis. PD contributed to the research ideas, interpretation of results and manuscript writing.

## Competing interests

The authors have no competing interests to declare.

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

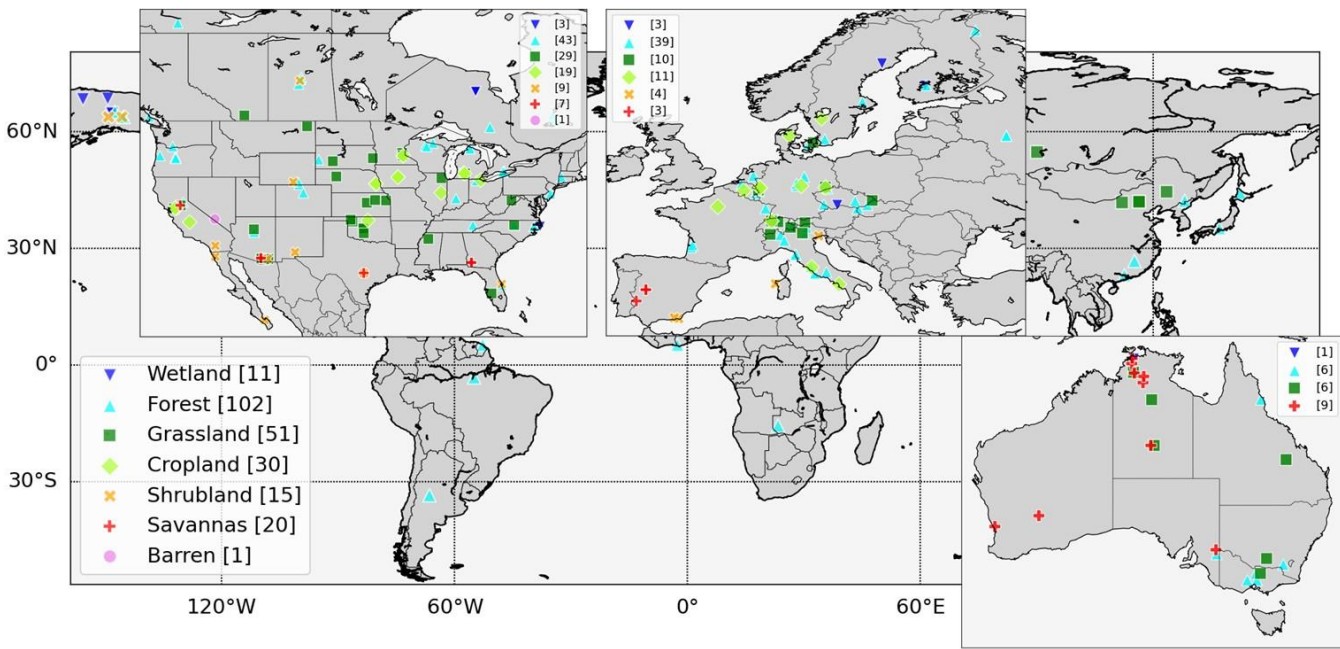

**Figure 1: Locations of flux sites marked according to reported IGBP land cover. The bracketed numbers indicate the number of sites reporting each corresponding land cover.**


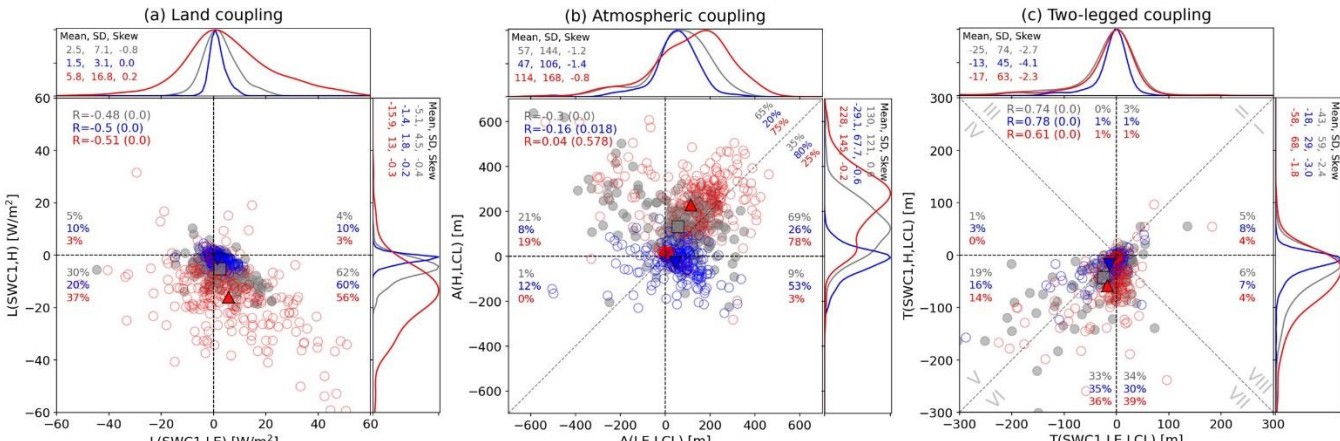

**Figure 2: Scatter plot of (a) land coupling, (b) atmospheric coupling, and (c) two-legged coupling for daily (grey), midday averaged (11–13 LST; red) and midnight averaged (23–01 LST; blue) values at 230 flux sites. Squares, upward triangles, and downward triangles indicate the mean across 230 sites for daily, midday, and midnight, respectively. Correlations and corresponding p-values (bracketed) are denoted in the upper-left corners. On each scatter diagram, percentages of stations in each quadrant (each octant for panel c) are indicated for daily, midday, and midnight data with corresponding colours. For the atmospheric coupling, percentages are also indicated on either side of the diagonal (y=x) line. The distribution of the kernel density estimations corresponding to x- and y-axis is shown as marginal distributions along the upper and right sides of each scatter plot. Each is normalized to have the same maximum value; the mean, standard deviation, and skewness for each distribution are also shown.**

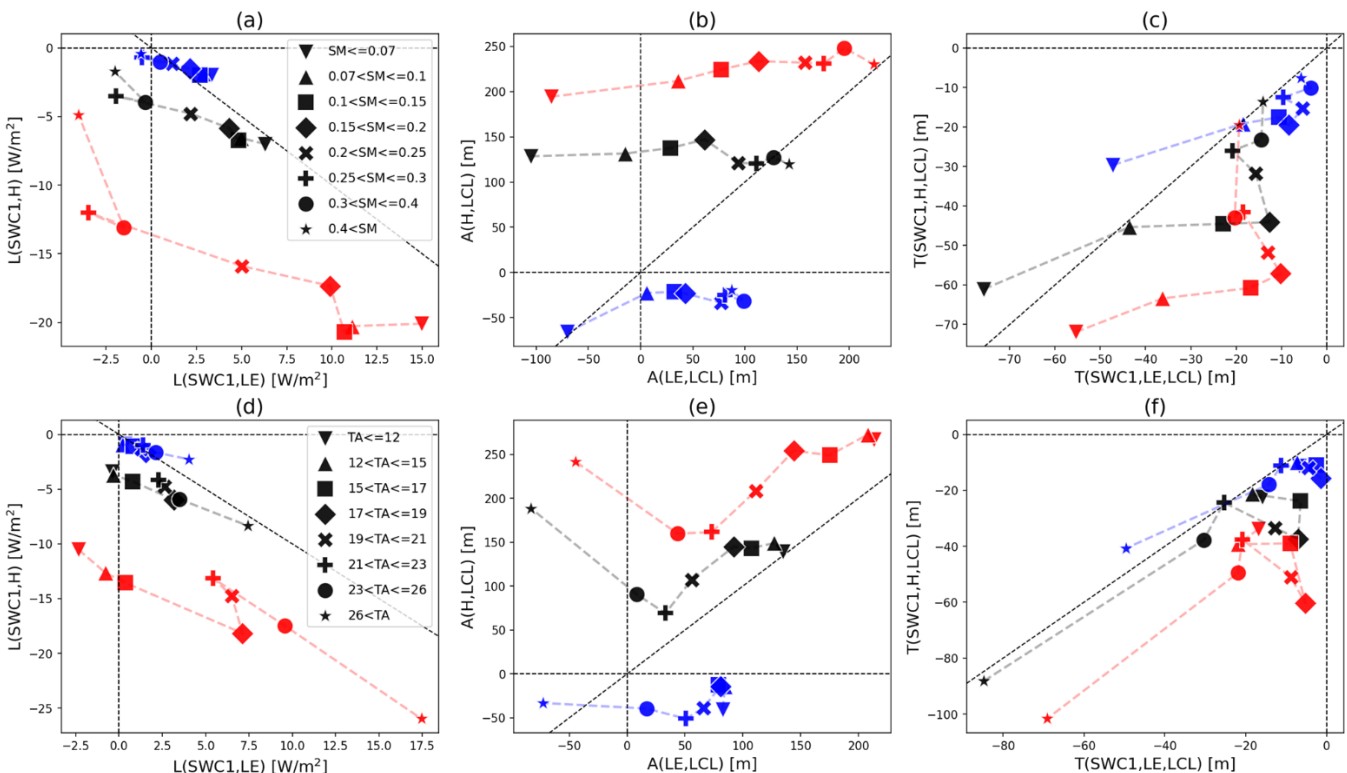

**Figure 3: Scatter plots between moisture (x-axes) and energy (y-axes) pathway couplings for (a, d) land, (b, e) atmospheric, and (c, f) two-legged coupling for daily mean (black), midday mean (11–13 LST; red) and midnight mean (23–01 LST; blue), composited by surface soil moisture (upper row) and surface air temperature (bottom row) ranges indicated by symbols in the legends. Ranges are chosen so that each category has a similar sample size. Values in adjacent ranges are connected by dashed lines.**

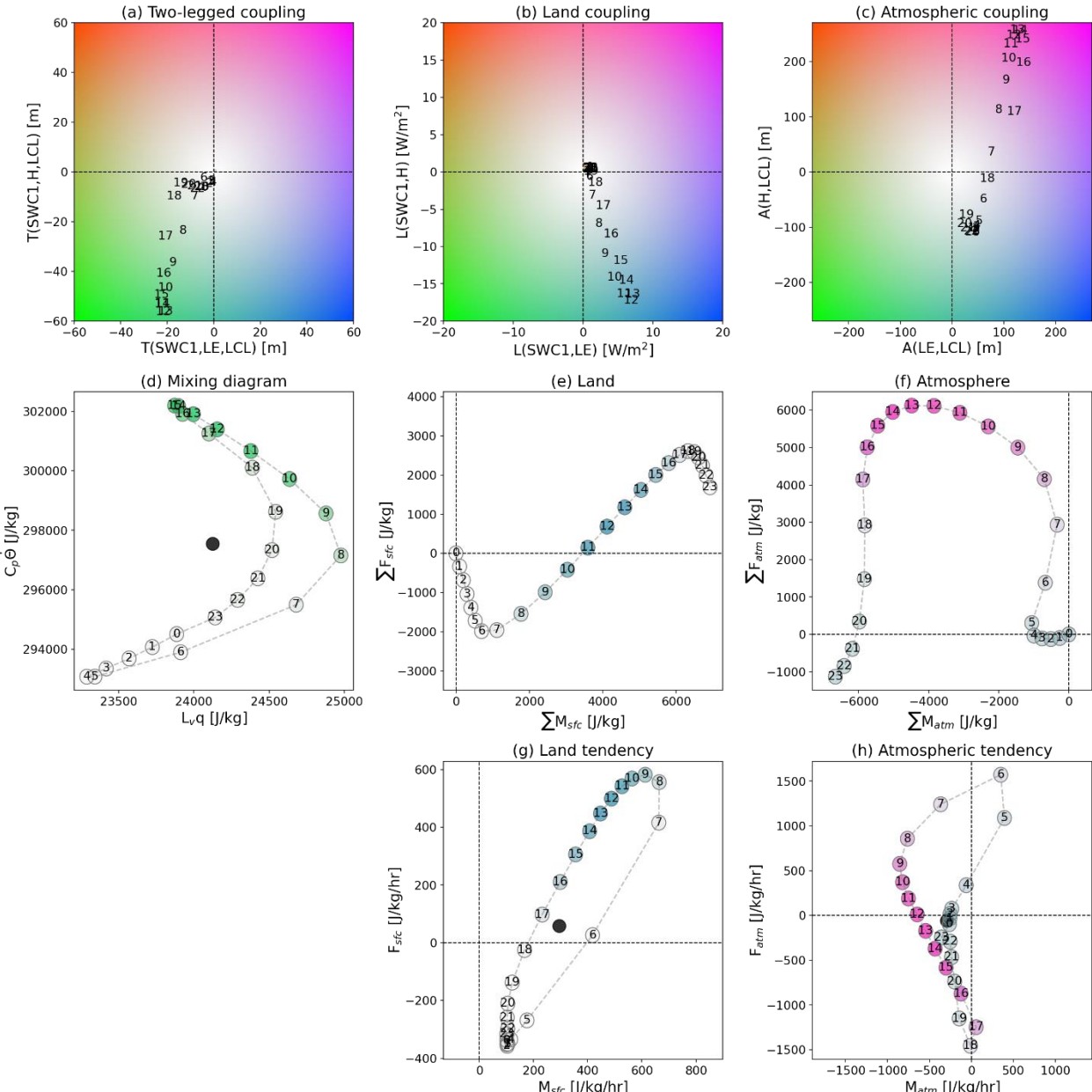

**Figure 4: Scatter plot of hourly (a) two-legged, (b) land, and (c) atmospheric couplings composed to LE- (x-axis) and H- (y-axis) relevant term in which the numbers indicate local hour. Shaded colours depend on the sign of LE- and H-related couplings such as green (LE[-], H[-]), blue (LE[+], H[-]), red (LE[-], H[+]), and purple (LE[+], H[+]), and colour saturation denotes the coupling strength. (d) The hourly mixing diagram plots moist (x-axis) and heat (y-axis) energy content per unit mass within the mixed layer. The circles are shaded by the colour determined by two-legged couplings in (a) corresponded to the local hour. The black circle is the mean of the 24-hourly values. (e) The land and (f) atmospheric components of diurnal mixing diagram, which represents the accumulated budgets relative to their corresponding vectors across the entire day, are shaded by land and atmospheric couplings, respectively. (g) The hourly land and (h) atmospheric vector representing their tendencies of the moist and heat energy budgets and the circles are shaded by corresponding couplings. In (g) and (h), the number represents the start of the hour over which tendencies are calculated.**

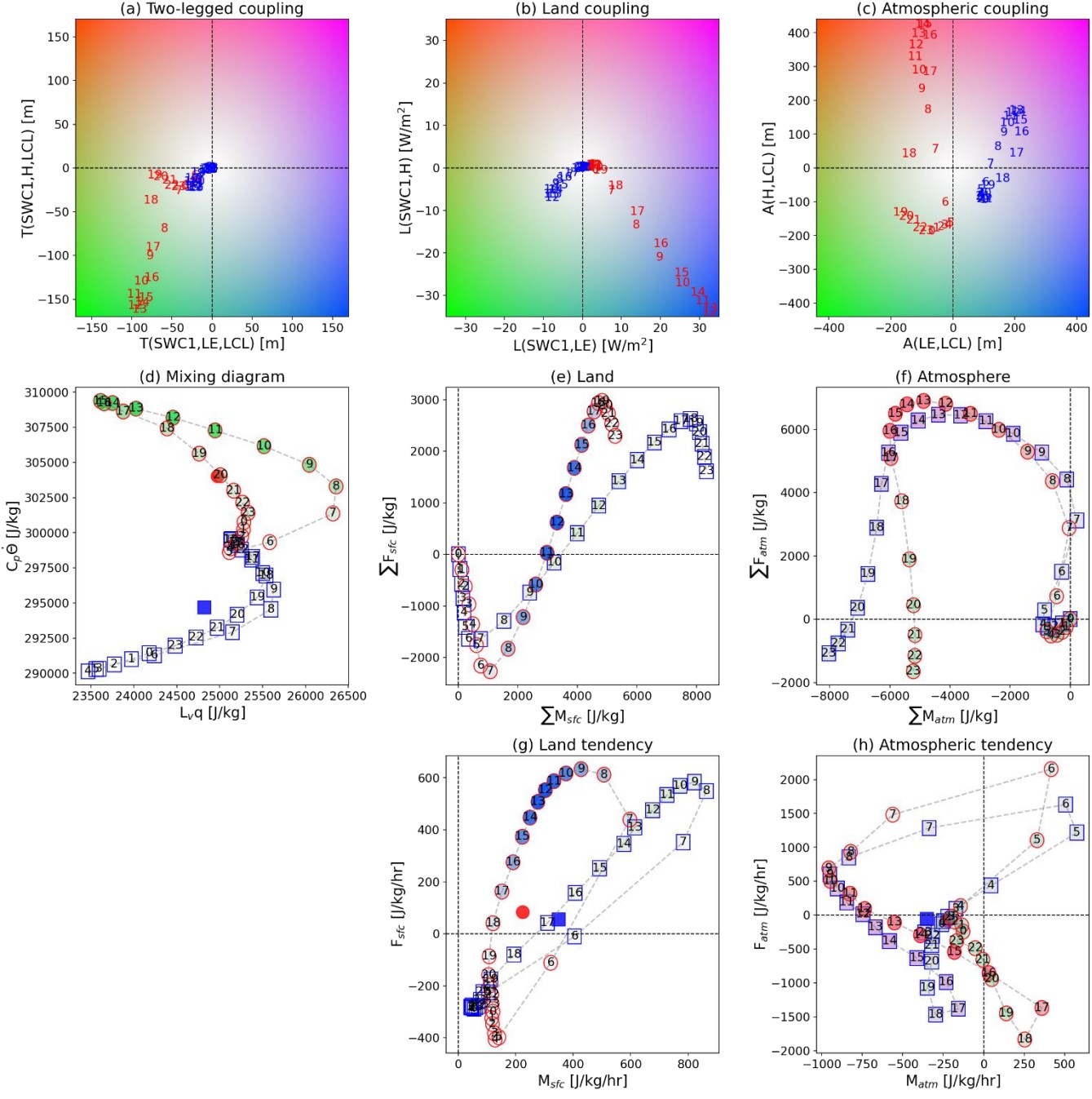

**Figure 5: Same as Fig. 4, but for the sensitivity of water-limited (circles outlined in red) and energy-limited (squares outlined in blue) regimes sampled by upper and lower 10% sites (N=23) as described in section 3.3.**

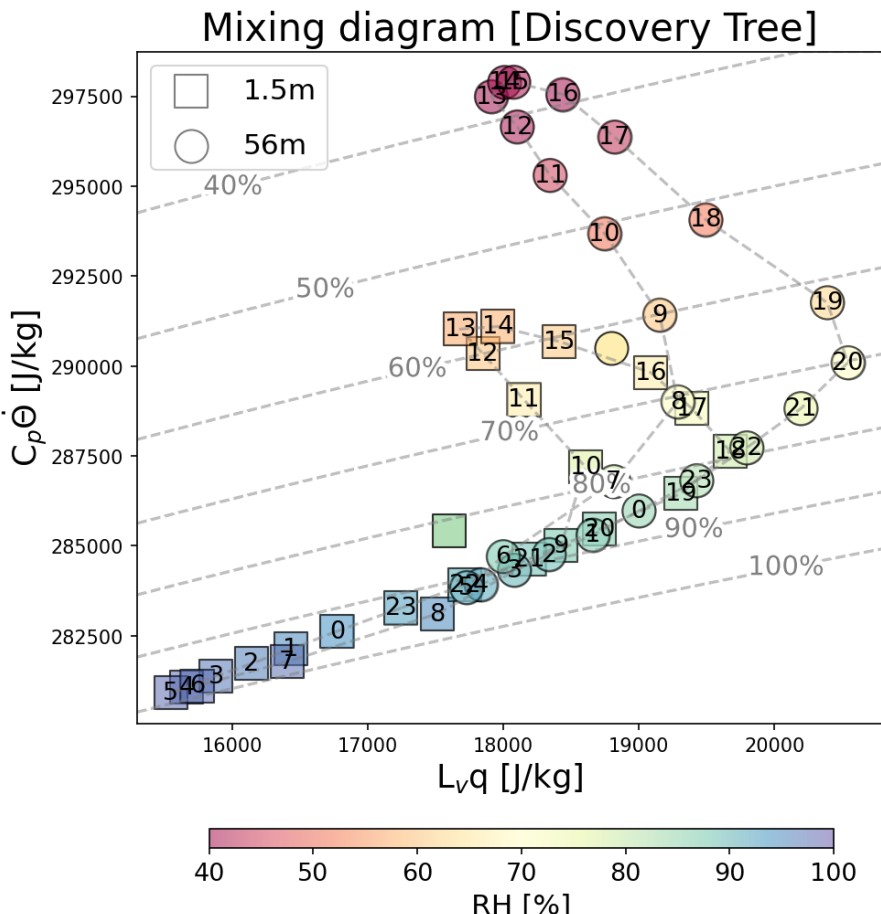

**Figure 6: Same as in Fig. 4d, but for the 1.5-m (squares) and 56-m (circles) height sensors at the Discovery Tree site, to examine the sensitivity of canopy microclimate physics on mixing diagrams that may affect the interpretation of forest flux tower sites. Grey dashed lines indicates the atmospheric relative humidity corresponding to the moisture (x-axis) and heat (y-axis) content. The shading indicates the climatological relative humidity corresponding to different hours.**