# Peer review of "Understanding the diurnal cycle of land-atmosphere interactions from flux site observations"

_Hydrology and Earth System Sciences, 2022_

## Referee Comment (RC1)

**Peer review of "Understanding the diurnal cycle of land-atmospheric interactions from flux-site observations"**
By: Raquel González Armas and Jordi Vilà-Guerau de Arellano

In this paper, land-atmospheric interactions are investigated based on observations of surface fluxes in 230 vegetated sites during the warm season (Figure 1). The overarching goal is the characterization of the main sub-daily features of the land-atmospheric couplings worldwide. The fluxes sites were part of the networks FLUXNET2015, AmeriFlux and European Drought-2018. The sites are classified as 7 different land types: (1) Wetlands, (2) Forest, (3) Grassland, (4) Cropland, (5) Shrubland, (6) Savannas and (7) Barren. The variables in which this study is based upon are: (1) soil water content in the top soil layer, (2) sensible heat flux, (3) latent heat flux, (4) surface air temperature, (4) humidity, (5) surface pressure and (6) vapor pressure deficit.

To investigate the land-atmospheric interactions two tools were mainly used: (1) the terrestrial coupling index and (2) mixing diagrams. The terrestrial coupling index is applied to both the land and the atmosphere. Furthermore, it is applied a third time in an integrative way to both land and atmosphere. In the paper, it is also applied a methodology to separate water and energy-limited regions based on the correlation of soil water content and the evaporative fraction. This separation makes possible to evaluate the couplings in each of both regimes.

Some interesting patterns are found. For instance, the fact that the two-legged coupling through the pathway of sensible heat is generally larger than that of latent heat at the sub-diurnal scale. This fact is obscured when considering the net diurnal contribution in which the latent heat coupling is stronger. Another relevant scientific acquisition is the corroboration with an extended data that the evening path of evaporation is characterized by a different behavior than the morning path.

In our opinion, the key strength of the research is the novelty of providing a first in-depth analysis of the sub-diurnal scale of the land-atmospheric coupling indexes. Since, these couplings are being used in climatological research, we find necessary to investigate their diurnal variability, and more specifically the sub-diurnal scales, and the sensitivity in these scales due to the clear diurnal behavior of the land-atmosphere interactions.

The manuscript lay down very well the actual state of art with the land-atmosphere interactions from a climatological point of view. It provides relevant past and ongoing research in the introduction. In addition, the methodology is well-explained and documented. In our opinion, the addition of the two-legged coupling which was proposed by Dirmeyer et. al. (2014) adds to the analysis. Even though this two-legged coupling is proportional to both the land and atmospheric couplings, when visualized for a large dataset it can uncover patterns not so easily identifiable by investigating the land and atmospheric coupling in isolation. Finally, we highly appreciate the scientific transparency. Developed code is available in GitHub.

We propose a major revision of the paper that we think could enrich the quality manuscript. Nonetheless, we find the publication valuable to the scientific community and of interest to the journal "Hydrology and Earth Systems".

Find below our main points:

**Major comments**

1.- Although the processes of entrainment and boundary layer growth is acknowledged throughout the paper, we have the feeling that is played down in the research. We realized that with a surface data set is difficult to quantify, although the mixed-layer diagrams proposed by Santanello et al. (2009) could be an adequate tool to further quantify the relevance of entrainment of warm and dry air at the different sites. Could the authors elaborate and quantify more regarding the role of entrainment?

2.- Closely connected to this, we miss key references at the introduction that can help the reader to position this research with respect to research that have already dealt with the relevance of processes happening at the sub-daily scales. For instance, Ek et al. (J. Hydro meteorology. **5**, 86–99, 2004) and van Heerwaarden et al (Quarterly Journal of the Royal Meteorological Society 135, 1277-1291, 2009). Could the authors introduce these or similar references in the introduction?

3.- At section 2.2 a key assumption is the use of the ERA5 to get information on the planetary boundary layer. Here, we disagree with the authors that the mesoscale variability of the boundary layer height is small compared to its temporal variability. There are clear examples in which the surface fluxes are not representative of the boundary layer development (see for instance figures 1 and 16 at Vilà-Guerau de Arellano et al (Biogeosciences 17, 2022). In that respect, there are already tools that enable us to make use of the worldwide soundings to determine the properties of the boundary layer dynamics (see Figures 3 and 4 at Hendricks et al., Geoscience Model Development 12, 2019). Although we realize that the use of this data set is beyond the scope of the paper, I believe the reader will appreciate a more elaborated and thorough sensitivity analysis on the uncertainty of ERA5 with respect to surface heterogeneity below the horizontal grid size of 31 km.

4.- Equation (5) describes how the pressure at the planetary boundary layer height was calculated by integrating equation (4). In doing so, the temperature must be expressed as a function of height. Assuming a linear dependency with height the following equation is reached.

$$P_{PBL} = P_{sfc} e^{-\frac{g\,(z_{PBL}-2)}{R\,(T_{PBL}-T_{2m})} ln\frac{T_{PBL}}{T_{2m}}}$$

In equation (5) a factor is missed inside the exponential. This factor is the inverse of the temperature lapse rate in the boundary layer, $\Gamma = \frac{T_{PBL}-T_{2m}}{z_{PBL}-2}$. In addition, in equation (5) the exponent has units when it must be dimensionless. I highly recommend correcting for this factor or for a similar corresponding factor if other assumptions were made.

5.- Along the results section in part *4.2 Diurnal mixing diagrams* and *4.3 Climate regime dependence*, hysteresis of the thermal process chain versus the moist process chain is discussed. Regarding the discussion of hysteresis, we have three comments:
1. We highly encourage to define in this context the term hysteresis. Hysteresis is a word originally coined in science to describe systems which state depends on their history. The typical scientific example is the magnetic hysteresis. This refers to a magnet that is able to experience different magnetic moments when subject to the same magnetic field. Those magnetic moments depend on the previous states of the magnet. To us, using hysteresis in land atmospheric context may be misleading since the state of the system may be different between morning and afternoon because the external factors are also different. For instance, soil water content and vapor pressure deficit are generally different between morning and afternoon. Therefore, the sub-diurnal asymmetry may be attributed to it not because an inherent change on the interactions due to the previous history. Nonetheless, we acknowledge that hysteresis term is generally used in land-atmospheric interactions context. We recommend defining the term in this context. We already find a definition in conclusions section, line 417, the fact that "the evening path through the water-energy phase does not retrace the morning path". We would move or repeat the definition to results because there is where the hysteresis is widely discussed. In addition, we think it would be valuable to specify in which way we consider it a hysteresis. In essence, which system is subject to its previous history? Is it the vegetation, is it a vegetation-soil system? What are considered the external factors? Another simpler solution is to coin another term such as temporal asymmetry which does not imply previous history relations.
2. We highly recommend discussing the hysteresis' possible causes both on the land and the atmospheric coupling. We argue that due to many processes that peak at different times (e.g., radiation peaks around noon, sensible heat flux peaks in the early afternoon and latent heat flux which with peaks later in the afternoon), morning-afternoon asymmetry can be expected.

It is not clear to us what is the added value of assessing the asymmetry or if the aim of the research is simply to characterize it. We recommend clarifying either if the paper aims to characterize them as a general characteristic observed or if the asymmetry is seen as a possible option to evaluate land atmosphere interactions.

**Omissions**

*Code not available yet in GitHub*
We just wanted to mention that the code is not yet available in the mentioned Github website. We guess that this may be made available after the publication. We just wanted to mention it to be sure that that was the case.

**Minor comments and typos**

*Comments about the introduction*
- We recommend reinforcing the importance of sub-diurnal variability to understand land-atmosphere interactions at longer time scales (e.g., seasonal, and climatological). In the paragraph of the introduction that goes from *line 55* to *line 68*, some examples are given. For instance, it is mentioned that it has been found links between morning evaporation and probability of rainfall, and between morning convective inhibition and convective initiation. If possible, we find interesting to include some more examples.
- The next paragraph that goes from *line 69* to *line 80* states that "… thorough examinations of complete diurnal cycle of land-atmospheric interactions have been lacking". We recommend clarifying that this is the case from the climatological point of view. Detailed study cases in which the diurnal cycle of land-atmospheric interactions is researched have been previously published. What we find relevant and innovative in this research is that the thorough sub-diurnal analysis focus on the coupling terms using long temporal time series spanning from 1996 to 2020. This climatological approach may reveal more generalizable land-atmospheric interactions.

*Comments on the methodology*
- *In line 109,* the lifting condensation level is used as the variable to understand the coupling of the land with the atmosphere. We think the reader would appreciate a short sentence in which it is stated why this variable is an important indicator of the coupling to the atmosphere (e.g., because its strong relation with cloud initiation or its importance in convection schemes in atmospheric models).
- *3.3 Mixing diagrams* section. Along this section mixing diagrams are introduced. It is stated that for computing them, 2-m temperature and humidity or vapor pressure deficit are used. In the last paragraph of the section, some shortcomings of this approach are addressed. For instance, it is mentioned that embedded in this method it lies one hypothesis. The hypothesis that 2-m measurements reflect mixed-layer values. We find this hypothesis to be dubious for certain ecosystems. For instance, in vegetated areas whose trees are taller than 2-m, the measurements fall into the in-canopy range. Many forests have trees that surpasses this height. Therefore, unlike many of the observations in other land types, observations in forests lie inside the canopy. In the research 102 from 230 sites (approx., 44 %) are classified as forests. Consequently, for forests sites, we wonder how much sensitive the land and surface couplings are to the height in which the surface heat fluxes, temperature and humidity are measured. We would expect that using measurements located right above the canopy would reflect different land and atmospheric coupling. We do acknowledge the challenge of comparing the diverse land-types considered in the study within the same methodological framework. Nonetheless, we would appreciate a justification of using the 2-m height measurements for forests or at least addressing the special advantages and shortcomings of such approach for forests. In addition, we wonder how the inclusion of these observations

affect the general conclusions for the land-atmospheric interactions. For instance, are patterns more easily generalizable (in figures 2, 3, 4 and 5) when forests are excluded?

- *Equations (6 and 8), page 7.* We find misleading the notation $H_{sfc}$ and $H_{atm}$ to term the hourly land and atmospheric vector component. H is generally used to depict a sensible heat surface flux. Therefore, in our opinion, a subindex to it would be a logical notation to indicate a partitioning of the flux. Nonetheless, in the notation used in the manuscript, the subindex is not indicating a partitioning of the flux itself but a partitioning of a slightly different variable. In this case instead of being a flux of energy per square meter (such H), $H_{sfc}$ and $H_{atm}$ refer to the amount of energy contained in a kilogram of air that has been introduced in a certain time (in this case one hour) due to either surface of atmospheric processes. Since the units and the physical variable are different, we recommend finding another symbol such M was used for the moisture vector components.

*Example when introducing terrestrial coupling*
*p5, line 137.* If the terrestrial *coupling* index is being introduced, it could be applied for both the land and atmosphere. In the text, the example of the land coupling index is stated. In this case, the target variable are the surface fluxes and the source variable the soil moisture content. Because this index will also be applied to the atmospheric coupling index, we suggest to propose the examples later or to substitute "i.e.," for "e.g.," when proposing the land coupling index as an application of the terrestrial coupling index.

*Asymmetry of L(SWC1,H)*
*p9, line 241* "This means that the asymmetry of L(SWC1,H) in the sub-daily time scale is larger than that of L(SWC1,LE), a characteristic that is explored in more detail later" we see that this is mentioned afterwards, but we recommend to indicate already here what processes may be affecting this asymmetry. These processes seem to be mostly diurnal. We think some interpretation of the physical processes, when possible, in the results may be enriching.

*Definition of significant relationships*
*p9, line 244* "The relationship between A(H,LCL) and A(LE,LCL) is not significant during midday due to their opposite relationships on either side of A(LE,LCL) = 0" What is specifically meant by "not significant"? It can be identified the two peaked distribution of A(LE,LCL) with one peak more predominantly in the region A(LE, LCL) > 0 and another less predominant in the region A(LE, LCL) < 0. What is the specific criteria to classify as "no significant"? Is it the fact that two feedbacks are identifiable? We would consider clarifying this point.

*Strength of the couplings*
*p9, line 256* Referring to figure 2c "Points on the right of the diagonal x=y line indicate stronger two-legged coupling through LE than trough H, which arise mainly from the larger correlation terms of land and atmosphere coupling via LE."
It is true that the points on the right of the diagonal y = x indicate that T(SWC1,LE,LCL) > T(SWC1, H, LCL). Nonetheless, in our opinion, that does not immediately mean that the two legged coupling is stronger because the coupling can be either positive (meaning a correlation between SWC1 and LCL through that pathway) or negative (meaning an uncorrelation between SWC1 and LCL through that pathway). To me, what indicates the strength of the coupling is the absolute value, that is: |T(SWC1,LE,LCL)| > |T(SWC1,H,LCL)|.
We have inserted a figure where the four regions that arise when the absolute values are considered are colored. Following that logic, the coupling following the LE pathway would be stronger for the regions II and IV. On the contrary, the coupling following the H pathway would be stronger for the regions I and III.  In that case, by naked eye, the strength of both couplings seems comparable, and it depends mainly on the density of points in regions I and IV. We would even argue that probably the coupling via the sensible heat flux pathway is stronger because in figure 4a, all values of the part where both couplings are negative are in the half corresponding to region I. In fact, in lines 292-294 it is accurately described this by stating: "During the daytime, both two-legged couplings are negative,

with T(SWC1, H, LCL) being almost three times as strong as T(SWC1,LE,LCL) around midday, showing the importance of sensible heating for ML growth".

---

## Author Comment (AC1)

*Many thanks for handling the review process for our manuscript. The time and effort devoted to our manuscript by you and the reviewers are very much appreciated.*

*We have revised the manuscript carefully according to the reviewers' comments and suggestions. In the following, we provide a point-by-point response. The original reviewer comments are in black regular font. Our responses are shown in blue italic font. Quotes from the revised paper are shown in blue bold-face font. Additionally, there are a number of small grammatical and wording changes throughout the manuscript that are not specifically documented below.*

REVIEWER COMMENTS

**Reviewer (Raquel González Armas and Jordi Vilà-Guerau de Arellano)**:

1. Although the processes of entrainment and boundary layer growth is acknowledged throughout the paper, we have the feeling that is played down in the research. We realized that with a surface data set is difficult to quantify, although the mixed-layer diagrams proposed by Santanello et al. (2009) could be an adequate tool to further quantify the relevance of entrainment of warm and dry air at the different sites. Could the authors elaborate and quantify more regarding the role of entrainment?

   ➔ *The role of atmospheric entrainment in the mixing diagrams, especially for the atmospheric components, introduces drier air with higher potential temperature at the top of the growing boundary layer. The impact is strongest from 7 AM to about noon while the boundary layer is growing. Its contribution is diminished during the afternoon since boundary layer growth slows and radiative cooling of the warmer air leads to a negative tendency of thermal energy, which becomes dominant at night. On the other hand, the sensitivity is not very different over the water- and energy-limited regimes despite the large discrepancy in the boundary layer depth. These descriptions is added in Lines 521-523 and 541-543:*

   **"Although the effect of atmospheric entrainment continues until continues until dissipation of the daytime boundary layer around sunset, it is obscured by the other contributions after noon."**

   **"Despite the PBL in water–limited regimes being about twice as deep as for energy-limited regimes, with accompanying stronger entrainment, the impact of the atmospheric entrainment over both climate regimes is similar, resulting in a positive temperature and a negative moisture tendency from 7 AM to the noon."**

2. Closely connected to this, we miss key references at the introduction that can help the reader to position this research with respect to research that have already dealt with the relevance of processes happening at the sub-daily scales. For instance, Ek et al. (J. Hydro meteorology. 5, 86–99, 2004) and van Heerwaarden et al (Quarterly Journal of the Royal Meteorological Society 135, 1277-1291, 2009). Could the authors introduce these or similar references in the introduction?

   ➔ *We have added the suggested key references to address the impact of sub-daily land-atmosphere interactions on the boundary layer development using the coupled model simulation. Ek and Holtslag (2004) highlighted the influence of land surface condition (e.g., soil moisture) on the development of the boundary layer cloud and Van Heerwaarden et al. (2009) addressed the effect of dry-air entrainment on surface evaporation and the convective boundary layer. These previous studies commonly focused on the daytime land-atmosphere feedback mechanisms. This description is added in Lines 70-73:*

> *"In addition, the influence of soil moisture on the boundary layer cloud development has been demonstrated for the coupled L–A system with realistic daytime surface fluxes and atmospheric profiles (Ek and Holtslag, 2004) and the role of dry-air entrainment has been shown to enhance surface evaporation and induce a shallower convective boundary layer through daytime L–A feedbacks (Van Heerwaarden et al., 2009)."*

3. At section 2.2 a key assumption is the use of the ERA5 to get information on the planetary boundary layer. Here, we disagree with the authors that the mesoscale variability of the boundary layer height is small compared to its temporal variability. There are clear examples in which the surface fluxes are not representative of the boundary layer development (see for instance figures 1 and 16 at Vilà-Guerau de Arellano et al (Biogeosciences 17, 2022). In that respect, there are already tools that enable us to make use of the worldwide soundings to determine the properties of the boundary layer dynamics (see Figures 3 and 4 at Hendricks et al., Geoscience Model Development 12, 2019). Although we realize that the use of this data set is beyond the scope of the paper, I believe the reader will appreciate a more elaborated and thorough sensitivity analysis on the uncertainty of ERA5 with respect to surface heterogeneity below the horizontal grid size of 31 km.

➔ *As the reviewer states, the surface fluxes reveal the sensitivity to spatial representativeness (Vilà-Guerau de Arellano et al. Biogeosciences 17, 2020), but the boundary layer height is relatively less sensitive to that problem (see for instance Fig. 16b in the same reference). Our study employs time series of the surface fluxes from FLUXNET2015 and the other flux sites, so that we only focus on the spatial representative issue in the PBL height as a function of time. To justify the adaptation of ERA5 PBL height, we have added some references that describe the limited spatial dependency of the modeled boundary layer in the CloudRoots field experiment, and that ERA5 is the best available reanalysis product among four different reanalysis datasets against worldwide radiosonde measurements. This description is added in Lines 145-149:*

> *"Although there are some issues in downscaling the gridded data to the observed sites due to unresolved spatial heterogeneity in the atmospheric boundary layer, Vilà-Guerau De Arellano et al. (2020) found a satisfactory agreement between ERA5 and three independent observations, which demonstrates that the boundary layer shows similar temporal evolution on the larger regional scale. Additionally, the inter-comparison of daytime $Z_{PBL}$ from four reanalysis datasets against globally distributed high-resolution radiosonde measurements suggests that the most accurate reanalysis product is ERA5 (Guo et al., 2021)."*

4. Equation (5) describes how the pressure at the planetary boundary layer height was calculated by integrating equation (4). In doing so, the temperature must be expressed as a function of height. Assuming a linear dependency with height the following equation is reached.

$$P_{PBL} = P_{sfc} e^{-\frac{g(Z_{PBL}-2)}{R(T_{PBL}-T_{2m})}\ln\frac{T_{PBL}}{T_{2m}}}$$

In equation (5) a factor is missed inside the exponential. This factor is the inverse of the temperature lapse rate in the boundary layer, $\Gamma = \frac{T_{PBL}-T_{2m}}{Z_{PBL}-2}$. In addition, in equation (5) the exponent has units when it must be dimensionless. I highly recommend correcting for this factor or for a similar corresponding factor if other assumptions were made.

➔ *Thank you for pointing out the missing term in the definition of pressure at PBL based on the vertical pressure gradient and the ideal gas law. Based on the reviewer's comments, we have*

*corrected the formulation of Eq. 5 and redone all the relevant calculations using the pressure at the PBL. Although the updated results are similar to the previous results, this has resulted in some noticeable but unsubstantial changes to Figs. 4 and 5.*

5. Along the results section in part 4.2 Diurnal mixing diagrams and 4.3 Climate regime dependence, hysteresis of the thermal process chain versus the moist process chain is discussed. Regarding the discussion of hysteresis, we have three comments:

   A. We highly encourage to define in this context the term hysteresis. Hysteresis is a word originally coined in science to describe systems which state depends on their history. The typical scientific example is the magnetic hysteresis. This refers to a magnet that is able to experience different magnetic moments when subject to the same magnetic field. Those magnetic moments depend on the previous states of the magnet. To us, using hysteresis in land atmospheric context may be misleading since the state of the system may be different between morning and afternoon because the external factors are also different. For instance, soil water content and vapor pressure deficit are generally different between morning and afternoon. Therefore, the sub-diurnal asymmetry may be attributed to it not because an inherent change on the interactions due to the previous history. Nonetheless, we acknowledge that hysteresis term is generally used in land-atmospheric interactions context. We recommend defining the term in this context. We already find a definition in conclusions section, line 417, the fact that "the evening path through the water-energy phase does not retrace the morning path". We would move or repeat the definition to results because there is where the hysteresis is widely discussed. In addition, we think it would be valuable to specify in which way we consider it a hysteresis. In essence, which system is subject to its previous history? Is it the vegetation, is it a vegetation-soil system? What are considered the external factors? Another simpler solution is to coin another term such as temporal asymmetry which does not imply previous history relations.

   ➔ *As reviewer recommends defining the hysteresis in the manuscript, we have moved its definition from the conclusion section to section 4.2, and clarify it in the Abstract as well. Before suggesting the result of the hysteresis defined by the diurnal cycle of mixing diagrams and L-A coupling, we describe that our definition of hysteresis indicates the temporal asymmetry during the entire diurnal cycle. This description is added as a modification to Lines 330-333:*

   **"There is a kind of hysteresis across the diurnal cycle when the terms are plotted this way, in that the thermal process chain leads the moist process chain by 2-3 hours. As a result, the evening path through the water-energy phase space does not retrace the morning path. The thermal coupling collapses toward zero quickly in the late afternoon, while the moist coupling declines gradually throughout the evening."**

   B. We highly recommend discussing the hysteresis' possible causes both on the land and the atmospheric coupling. We argue that due to many processes that peak at different times (e.g., radiation peaks around noon, sensible heat flux peaks in the early afternoon and latent heat flux which with peaks later in the afternoon), morning-afternoon asymmetry can be expected. It is not clear to us what is the added value of assessing the asymmetry or if the aim of the research is simply to characterize it. We recommend clarifying either if the paper aims to characterize them as a general characteristic observed or if the asymmetry is seen as a possible option to evaluate land atmosphere interactions.

   ➔ *We did not deeply discuss the hysteresis in land and atmospheric couplings in Figs. 4b and 4c. Due to the lack of the description about the possible causes of those morning-afternoon*

*asymmetry, it was difficult to assess what leads to the diurnal hysteresis in land and atmospheric couplings. To characterize the observed diurnal coupling behaviors, this description is added in Lines 337-339 and 344-346:*

***"This results from the asymmetry of R(SWC1,LE) between developing (morning) and decaying (afternoon) phases, whereas R(SWC1,H) is relatively symmetric (not shown)."***

***"The diurnal atmospheric coupling hysteresis is determined by the evolution of R(H,LCL) and R(LE,LCL) and emphasized by the large daytime LCL variability. It is characterized by the diurnal maximum of LCL variance at 3-4 PM and its abrupt decaying from that maximum."***

6. Omissions - Code not available yet in GitHub

   We just wanted to mention that the code is not yet available in the mentioned Github website. We guess that this may be made available after the publication. We just wanted to mention it to be sure that that was the case.

   ➔ *We have posted the source code used in these calculations and the plots included in this manuscript through GitHub. They had been left in private mode after submission – they are now publicly available.*

7. Comments about the introduction

   A. We recommend reinforcing the importance of sub-diurnal variability to understand land-atmosphere interactions at longer time scales (e.g., seasonal, and climatological). In the paragraph of the introduction that goes from line 55 to line 68, some examples are given. For instance, it is mentioned that it has been found links between morning evaporation and probability of rainfall, and between morning convective inhibition and convective initiation. If possible, we find interesting to include some more examples.

   ➔ *We omitted the specified season of the previous studies in the introduction section. Most of the previous studies examined the climatological land-atmosphere interactions and they tend to focus on the summer season when the land-atmosphere coupling is prominent. The specific description is added in Lines 63-64 and 67-69:*

   ***"Findell et al. (2011) established that increased morning evaporation leads to an enhanced probability of afternoon rainfall for the boreal summer season over much of the United States, …"***

   ***"The climatological probability of summertime convective initiation was found to be more sensitive to morning convective inhibition over the southeastern United States, …"***

   B. The next paragraph that goes from line 69 to line 80 states that "… thorough examinations of complete diurnal cycle of land-atmospheric interactions have been lacking". We recommend clarifying that this is the case from the climatological point of view. Detailed study cases in which the diurnal cycle of land-atmospheric interactions is researched have been previously published. What we find relevant and innovative in this research is that the thorough sub-diurnal analysis focus on the coupling terms using long temporal time series spanning from 1996 to 2020. This climatological approach may reveal more generalizable land-atmospheric interactions.

➜ *The purpose of this study is highlighted in the last paragraph of the introduction section where we mention this study investigates the climatological land-atmosphere interactions using globally distributed flux tower observations. To clarify the lack of previous studies dealing with the full diurnal cycle of the climatological land-atmosphere interactions, we amended the wording in Line 74:*

**"Nevertheless, thorough examinations of the climatology of the complete diurnal cycle of L–A interactions have been lacking."**

8. Comments on the methodology

A. In line 109, the lifting condensation level is used as the variable to understand the coupling of the land with the atmosphere. We think the reader would appreciate a short sentence in which it is stated why this variable is an important indicator of the coupling to the atmosphere (e.g., because its strong relation with cloud initiation or its importance in convection schemes in atmospheric models).

➜ *This study employs the lifting condensation level to characterize the atmospheric behavior based on the readily-available near-surface atmospheric conditions. It is used to understand the atmospheric coupling and two-legged coupling from the flux tower observations. It also can be compared to the PBL height to understand cloud formation processes in terms of land-atmosphere interactions. This description is added in Lines 121-124:*

**"The LCL can be characterized as a potential level of cloud base formation based on parcel theory. It can be compared to the planetary boundary layer (PBL) height to define LCL deficit (PBL height minus LCL; Santanello et al., 2011). When the PBL grows to the height of the LCL (corresponding to positive values of the LCL deficit), water may condense from the air parcel and cloud formation occurs."**

B. 3.3 Mixing diagrams section. Along this section mixing diagrams are introduced. It is stated that for computing them, 2-m temperature and humidity or vapor pressure deficit are used. In the last paragraph of the section, some shortcomings of this approach are addressed. For instance, it is mentioned that embedded in this method it lies one hypothesis. The hypothesis that 2-m measurements reflect mixed-layer values. We find this hypothesis to be dubious for certain ecosystems. For instance, in vegetated areas whose trees are taller than 2-m, the measurements fall into the in-canopy range. Many forests have trees that surpasses this height. Therefore, unlike many of the observations in other land types, observations in forests lie inside the canopy. In the research 102 from 230 sites (approx., 44 %) are classified as forests. Consequently, for forests sites, we wonder how much sensitive the land and surface couplings are to the height in which the surface heat fluxes, temperature and humidity are measured. We would expect that using measurements located right above the canopy would reflect different land and atmospheric coupling. We do acknowledge the challenge of comparing the diverse land-types considered in the study within the same methodological framework. Nonetheless, we would appreciate a justification of using the 2-m height measurements for forests or at least addressing the special advantages and shortcomings of such approach for forests. In addition, we wonder how the inclusion of these observations affect the general conclusions for the land-atmospheric interactions. For instance, are patterns more easily generalizable (in figures 2, 3, 4 and 5) when forests are excluded?

➜ *Flux tower observations from FLUXNET2015 include the surface air temperature, humidity, air pressure, vapor pressure, surface fluxes above the canopy layer and sometimes within it.*

*However, the documentation generally does not provide the canopy information needed to compare it with the reference height of the measurements. Although this limitation is a shortcoming of the usage of these flux tower datasets, we have assumed the variables are measured from 2 meters above the canopy. So, we added this description for the flux measurements in Lines 116-120, 199, and 201-202:*

**"Except for SWC1, the other variables are assumed to have been measured a few meters above the canopy while acknowledging that the canopy height varies among sites. However, the flux observations generally do not contain the canopy information necessary to compare to the reference height of the sensors, which is a shortcoming of using flux tower data, especially for forested locations."**

**"Flux sites provide surface air temperature …"**

**"As the instrument height may vary among flux towers, this study assumes that the observations are taken 2 meters above the canopy."**

➔ *To understand the possible impact of sub-canopy measurements in the diurnal mixing diagram, we have additionally employed the meteorological data from the Discovery Tree at the Andrews Experimental Forest in Oregon, USA, which provides vertical temperature and humidity observation at 1.5- (below canopy) and 56-meter (above canopy). Although the result of the diurnal mixing diagrams is from only this single site in a forest, it can give some insight and necessity to understand not only the below- but also above-canopy physics. Thus, we add the new analysis (Fig. 6) and its description in section 2.1 (Lines 125-132), section 4.4 (Lines 426-449), and section 5 (Lines 533-541). It exposes the shortcomings of the standardized FLUXNET2015 dataset (and similar data, e.g., AmeriFlux) for such land-atmosphere studies, as there are clear contrasts between different heights within and above the canopy. The data we used represents an extreme case (in a very tall, dense, old growth forest) – perhaps other sites will show less contrast.*

➔ *To account for the contribution of the forest regions to the overall climatological result, we have reproduced the analysis in Fig. 3, but sorted by vegetation type. The result shows the forest does not reveal any distinctive characteristics in the comparison of segmented couplings across the types of vegetated land cover. For instance, wetlands generally agree with the results of wet and cold climate, and the savanna is consistent with the result of dry and warm climate. Of course, there is correlation between climate regime and the vegetation that can grow there.*

[Figure]

*The diurnal cycle of mixing diagrams and land-atmosphere couplings (Fig. 5) is also reproduced by separating forest sites (site count is 102 and indicated by blue squares) to the other (site count is 128 and indicated by red circles). The results show that the sensitivity of the forest land cover to the diurnal cycle of the land-atmosphere interaction is not clear, but the forest sites are placed mainly in the energy-limited regimes which indicates some sensitivity*

[Figure]

*to the background state. Again, the climate signal is necessarily convolved with vegetation.*

C. Equations (6 and (8), page 7. We find misleading the notation Hsfc and Hatm to term the hourly land and atmospheric vector component. H is generally used to depict a sensible heat surface flux. Therefore, in our opinion, a subindex to it would be a logical notation to indicate a partitioning of the flux. Nonetheless, in the notation used in the manuscript, the subindex is not indicating a partitioning of the flux itself but a partitioning of a slightly different variable. In this case instead of being a flux of energy per square meter (such H), Hsfc and Hatm refer to the amount of energy contained in a kilogram of air that has been introduced in a certain time (in this case one hour) due to either surface of atmospheric processes. Since the units and the physical variable are different, we recommend finding another symbol such M was used for the moisture vector components.

➔ *We replace the notation for the heat fluxes within the energy budget in the mixing diagrams with $F_{sfc}$ and $F_{atm}$ over the entire manuscript.*

9. Asymmetry of L(SWC1,H)

   p9, line 241 "This means that the asymmetry of L(SWC1,H) in the sub-daily time scale is larger than that of L(SWC1,LE), a characteristic that is explored in more detail later" we see that this is mentioned afterwards, but we recommend to indicate already here what processes may be affecting this asymmetry. These processes seem to be mostly diurnal. We think some interpretation of the physical processes, when possible, in the results may be enriching.

   ➔ *The diurnal sign shift of H is from positive during the daytime to negative during the nighttime. It is the major factor for the larger asymmetry of L(SWC1,H) in the sub-daily time scale compared with that of L(SWC1,LE). This description is added in Lines 268-269:*

   **"This means that the asymmetry of L(SWC1,H) in the sub-daily time scale is larger than that of L(SWC1,LE), which is mainly attributed to the diurnal reversal of H (positive during the day and negative at night). This characteristic is explored in more detail later."**

10. Definition of significant relationships

    p9, line 244 "The relationship between A(H,LCL) and A(LE,LCL) is not significant during midday due to their opposite relationships on either side of A(LE,LCL) = 0" What is specifically meant by "not significant"? It can be identified the two peaked distribution of A(LE,LCL) with one peak more predominantly in the region A(LE, LCL) > 0 and another less predominant in the region A(LE, LCL) < 0. What is the specific criteria to classify as "no significant"? Is it the fact that two feedbacks are identifiable? We would consider clarifying this point.

    ➔ *We meant to address that the p-value calculated from the correlation between the 230 sampled values of A(H,LCL) and A(LE,LCL) is large though "not significant". This is clarified in Lines 271-272:*

    **"The relationship between A(H,LCL) and A(LE,LCL) is not significant during midday, based on a high p-value along with low correlation, …"**

11. Strength of the couplings

    p9, line 256 Referring to figure 2c "Points on the right of the diagonal x=y line indicate stronger two-legged coupling through LE than trough H, which arise mainly from the larger correlation terms of land and atmosphere coupling via LE."

    It is true that the points on the right of the diagonal y = x indicate that T(SWC1,LE,LCL) > T(SWC1, H, LCL). Nonetheless, in our opinion, that does not immediately mean that the two legged coupling is stronger because the coupling can be either positive (meaning a correlation between SWC1 and LCL through that pathway) or negative (meaning an uncorrelation between SWC1 and LCL through that pathway). To me, what indicates the strength of the coupling is the absolute value, that is: |T(SWC1,LE,LCL)| > |T(SWC1,H,LCL)|.

    We have inserted a figure where the four regions that arise when the absolute values are considered are colored. Following that logic, the coupling following the LE pathway would be stronger for the regions II and IV. On the contrary, the coupling following the H pathway would be stronger for the regions I and III. In that case, by naked eye, the strength of both couplings seems comparable, and it depends mainly on the density of points in regions I and IV. We would even argue that probably the coupling via the sensible heat flux pathway is stronger because in figure 4a, all values of the

part where both couplings are negative are in the half corresponding to region I. In fact, in lines 292-294 it is accurately described this by stating: "During the daytime, both two-legged couplings are negative, with T(SWC1, H, LCL) being almost three times as strong as T(SWC1,LE,LCL) around midday, showing the importance of sensible heating for ML growth".

[Figure]

➔ *We have typically focused on the quadrant where both two-legged couplings are negative as the realm where land surface state variability directly affects the atmosphere. But, as the reviewer mentioned, the absolute comparison between both couplings can suggest the comprehensive interpretation for which mediative land fluxes highly contribute to LCL variability (or are at least related if not causal) regardless the coupling sign. Thus, we have modified the Fig. 2c by removing regression lines [which erroneously implied y=f(x)] and separating the quadrants into an 8-piece domain of triangles separated by y=x and y=-x lines as well as y=0 and x=0. The percentages denote the population in each octant. The updated figure is shown below and its description is in Lines 282-288 and Lines 323-324.*

[Figure]

*"The observed two-legged couplings from soil moisture to LCL, mediated by H and LE, are mostly placed on the left side of the y=-x line (Fig. 2c). Most couplings are negative, which means LCL height is anticorrelated with soil moisture regardless of the pathway of the coupling. To the left of the y=-x line (octants IV through VII), points in octants VI and VII indicate stronger two-legged coupling of soil moisture control on potential cloud base through H rather than through LE. Locations presenting stronger coupling through H are almost two times more than through LE throughout the entire day. This arises mainly from the larger correlation in the terms of land and atmosphere coupling via H; the LCL is less sensitive to LE variability compared with H, particularly in dry land conditions (not shown)."*

*"It is consistent with the result of three times more locations exhibiting stronger two-legged coupling of soil moisture to LCL through H than through LE (c.f., Fig. 2c)."*

---

## Author Comment (AC2)

*Many thanks for handling the review process for our manuscript. The time and effort devoted to our manuscript by you and the reviewers are very much appreciated.*

*We have revised the manuscript carefully according to the reviewers' comments and suggestions. In the following, we provide a point-by-point response. The original reviewer comments are in black regular font. Our responses are shown in blue italic font. Quotes from the revised paper are shown in blue bold-face font. Additionally, there are a number of small grammatical and wording changes throughout the manuscript that are not specifically documented below.*

REVIEWER COMMENTS

**Reviewer (Timothy Lahmers)**:

Major Comments:

1. Section 3.1: When analysis days were selected, major precipitation events were removed based on daily soil moisture tendencies. Is the 2-standard deviation threshold in soil moisture tendencies, for removing precipitation days from the analysis, sufficient? This method could still theoretically be affected by convection, especially in more arid environments where deep convection may occur even if rainfall is relatively light.

   ➔ *As many of the flux tower sites do not (reliably) report precipitation, the available observations are largely decreased if we use flux tower sites which commonly observe precipitation and other variables. Thus, major precipitation events were removed based on daily soil moisture tendencies and the 2-standard deviation is used for the threshold value. We tried masking out the major precipitation with different threshold values. 2-standard deviation works well to mask out the precipitation over various sites even though the precipitation may be characterized by convective features and other background states.*

2. Section 3.4: The authors selected a method to separate water and energy-limited environments using a correlation between soil water content and evaporative fraction. Could the authors provide prior literature or observation data to justify this method? Has this selection method been compared to other widely used proxies for aridity, such as the Budyko curve?

   ➔ *Contrasting EF sensitivities to soil moisture variations are evident in the energy-limited (zero slope) and water-limited (positive slope) regimes. Therefore, the correlation coefficient between daily EF and soil moisture can be identified by fitting a piece-wise linear function to the observed relationship between soil moisture and EF. Relevant references are added in the manuscript (Line 249)*

   - *Dirmeyer, P. A., Zeng, F. J., Ducharne, A., Morrill, J. C., and Koster, R. D.: The sensitivity of surface fluxes to soil water content in three land surface schemes, Journal of Hydrometeorology, 1, 121-134, 2000.*

   - *Dong, J., Akbar, R., Short Gianotti, D. J., Feldman, A. F, Crow, W. T., and Entekhabi, D.: Can Surface Soil Moisture Information Identify Evapotranspiration Regime Transitions?, Geophysical Research Letters, 49, e2021GL097697, 2022.*

3. Section 5: The findings of this paper are important for both the atmospheric modeling and observation applications of the PBL community. I would suggest that the conclusions include a

more substantial discussion of the implications of this work for future atmospheric model development, such as for PBL parameterizations in mesoscale models. Also, consider breaking section 5 up into two sections. Lines 400-478 are more of a summary, while lines 479-499 are more of a discussion about the significance and potential for future work based on these results. These new sections could be broken up accordingly.

➔ *Breaking section 5 into separate sections for conclusions and discussions seems to be better to isolate the context of this study and its implementation and novelty. Furthermore, the implementation of atmospheric modeling is added in Lines 554-556:*

     ***"This study is also of potential value for future atmospheric model development, such as for PBL and convective parameterizations in mesoscale models on a sub-daily time scale."***

Technical/Minor Comments:

4. The figures provide useful information to the readers; however, the labels and values shown on the x and y-axis are relatively small and difficult to read. Consider revising the figures to make key values for the reader more legible.

➔ *All the figures are modified to increase the label size for the reader more legible.*

---

## Referee Report (RR1)

**Second Peer review of**
**"Understanding the diurnal cycle of land-atmospheric interactions from flux-site observations"**
By: Raquel González Armas and Jordi Vilà-Guerau de Arellano

We thank the authors for their responses and the changes. In our opinion, the manuscript has improved.

Below our reaction to their responses and the changes in the manuscript (in blue). If there is no reaction, we are satisfied with their answers and the revised manuscript.

**Major comments**

1.- Although the processes of entrainment and boundary layer growth is acknowledged throughout the paper, we have the feeling that is played down in the research. We realized that with a surface data set is difficult to quantify, although the mixed-layer diagrams proposed by Santanello et al. (2009) could be an adequate tool to further quantify the relevance of entrainment of warm and dry air at the different sites. Could the authors elaborate and quantify more regarding the role of entrainment?

Could they be more precise on the projects dominant after midday when entrainment becomes less relevant? The word obscure is vague in the revised manuscript.

We also think that they are missing here a great opportunity in elaborating a bit more in the relevance of the processes at the sub-diurnal scales. Our recommendation is that based on their analysis and metrics they performed a more deep analysis on which processes are relevant or differ under the water and energy regimes.

Please rephrase and elaborate more (in red there is typo):

*"Although the effect of atmospheric entrainment continues until continues until dissipation of the daytime boundary layer around sunset, it is obscured by the other contributions after noon."*

5.- Along the results section in part *4.2 Diurnal mixing diagrams* and *4.3 Climate regime dependence,* hysteresis of the thermal process chain versus the moist process chain is discussed. Regarding the discussion of hysteresis, we have three comments:

1. We highly encourage to define in this context the term hysteresis. Hysteresis is a word originally coined in science to describe systems which state depends on their history. The typical scientific example is the magnetic hysteresis. This refers to a magnet that is able to experience different magnetic moments when subject to the same magnetic field. Those magnetic moments depend on the previous states of the magnet. To us, using hysteresis in land atmospheric context may be misleading since the state of the system may be different between morning and afternoon because the external factors are also different. For instance, soil water content and vapor pressure deficit are generally different between morning and afternoon. Therefore, the sub-diurnal asymmetry may be attributed to it not because an inherent change on the interactions due to the previous history. Nonetheless, we acknowledge that hysteresis term is generally used in land-atmospheric interactions context. We recommend defining the term in this context. We already find a definition in conclusions section, line 417, the

fact that "the evening path through the water-energy phase does not retrace the morning path". We would move or repeat the definition to results because there is where the hysteresis is widely discussed. In addition, we think it would be valuable to specify in which way we consider it a hysteresis. In essence, which system is subject to its previous history? Is it the vegetation, is it a vegetation-soil system? What are considered the external factors? Another simpler solution is to coin another term such as temporal asymmetry which does not imply previous history relations.

What do you mean by a kind of hysteresis? Is it not more appropriate to call it asymmetry? Please formulate with precision

2. We highly recommend discussing the hysteresis' possible causes both on the land and the atmospheric coupling. We argue that due to many processes that peak at different times (e.g., radiation peaks around noon, sensible heat flux peaks in the early afternoon and latent heat flux which with peaks later in the afternoon), morning-afternoon asymmetry can be expected. It is not clear to us what is the added value of assessing the asymmetry or if the aim of the research is simply to characterize it. We recommend clarifying either if the paper aims to characterize them as a general characteristic observed or if the asymmetry is seen as a possible option to evaluate land atmosphere interactions.

It is a pity that the authors do not develop and elaborate a bit more on the physics of this asymmetry based on the observational analysis

Other Comments

- *In line 109,* the lifting condensation level is used as the variable to understand the coupling of the land with the atmosphere. We think the reader would appreciate a short sentence in which it is stated why this variable is an important indicator of the coupling to the atmosphere (e.g., because its strong relation with cloud initiation or its importance in convection schemes in atmospheric models).

  Perhaps here it is convenient to be more rigorous and stress that the condition $h > LCL$ is a rough approximation. Majority of the situations in which shallow cumulus form are characterized by an opposite situation ($h < LCL$) (see for instance figure 7a at https://journals.ametsoc.org/view/journals/atsc/71/3/jas-d-13-0192.1.xml) . Could they please elaborate a bit more here?

- *3.3 Mixing diagrams* section. Along this section mixing diagrams are introduced. It is stated that for computing them, 2-m temperature and humidity or vapor pressure deficit are used. In the last paragraph of the section, some shortcomings of this approach are addressed. For instance, it is mentioned that embedded in this method it lies one hypothesis. The hypothesis that 2-m measurements reflect mixed-layer values. We find this hypothesis to be dubious for certain ecosystems. For instance, in vegetated areas whose trees are taller than 2-m, the measurements fall into the in-canopy range. Many forests have trees that surpasses this height. Therefore, unlike many of the observations in other land types, observations in forests lie inside the canopy. In the research 102 from 230 sites (approx., 44 %) are classified as forests. Consequently, for forests sites, we wonder how much sensitive the land and surface couplings are to the height in which the surface heat fluxes, temperature and humidity

are measured. We would expect that using measurements located right above the canopy would reflect different land and atmospheric coupling. We do acknowledge the challenge of comparing the diverse land-types considered in the study within the same methodological framework. Nonetheless, we would appreciate a justification of using the 2-m height measurements for forests or at least addressing the special advantages and shortcomings of such approach for forests. In addition, we wonder how the inclusion of these observations affect the general conclusions for the land-atmospheric interactions. For instance, are patterns more easily generalizable (in figures 2, 3, 4 and 5) when forests are excluded?

In the new manuscript it has been written that the observations of FLUXNET2015 data have been assumed to be taken at 2 m above the canopy. We do not fully understand what does "assume" means here. Does it mean that it is unknown the convention of height of FLUXNET2015? For instance, is it unknown if the documentation means 2m height from the surface or 2m height above the canopy? We would like some clarification or contacting FLUXNET for further details about the height of measurements with respect to height of the trees.

New comments connected to the revised manuscript

Regarding the addition of figure 6 and section "4.4. Canopy effects", we think it adds value to the manuscript. We think figure 6(b) could be omitted because the processes explained in the section can be visualized from figure 6(a), so we do not think it adds value to the analysis. In addition, it is a new way of visualization that has not been used before in the manuscript so it may disorientate the reader.

Section 5 is called "Conclusions" whereas section 6 is called "Discussion". We think it must be the other way around.

---

## Author Response (AR2)

*Many thanks for handling the review process for our manuscript. The time and effort devoted to our manuscript by you and the reviewers are very much appreciated.*

*We have revised the manuscript carefully according to the reviewers' comments and suggestions. This includes a number of minor wording changes throughout the document to improve clarity and readability. In the following, we provide a point-by-point response to the reviewers' comments. The original comments are in black and blue regular font. Our responses are shown in blue italic font. Quotes from the revised paper are shown in green bold-face font. Additionally, there are a number of small grammatical and wording changes throughout the manuscript that are not specifically documented below.*

REVIEWER COMMENTS

**Reviewer (Raquel González Armas and Jordi Vilà-Guerau de Arellano)**:

We thank the authors for their responses and the changes. In our opinion, the manuscript has improved.

Below our reaction to their responses and the changes in the manuscript (in blue). If there is no reaction, we are satisfied with their answers and the revised manuscript.

**Major comments**

1. Although the processes of entrainment and boundary layer growth is acknowledged throughout the paper, we have the feeling that is played down in the research. We realized that with a surface data set is difficult to quantify, although the mixed-layer diagrams proposed by Santanello et al. (2009) could be an adequate tool to further quantify the relevance of entrainment of warm and dry air at the different sites. Could the authors elaborate and quantify more regarding the role of entrainment?

   Could they be more precise on the projects dominant after midday when entrainment becomes less relevant? The word obscure is vague in the revised manuscript.

   We also think that they are missing here a great opportunity in elaborating a bit more in the relevance of the processes at the sub-diurnal scales. Our recommendation is that based on their analysis and metrics they performed a more deep analysis on which processes are relevant or differ under the water and energy regimes.

   ➔ *We have tried to elaborate the impact of the entrainment as well as radiative cooling in the Summary section. This is discussed in the last paragraph of section 4.2. To characterize further the atmospheric entrainment in terms of heat and moist energy budget, we refer to a previous study that investigated the boundary layer budgets using radiosonde data (Barr and Betts, 1997). This contributes to the range of the diurnal variance in moist and thermal energy budgets in the mixing diagram. Text is added in Lines 537-544:*

   **"The effect of atmospheric entrainment is greatest during the period of ML growth in the morning, when the entrained dry and high potential temperature air at the top of the PBL causes positive temperature and negative moisture tendencies in the ML. Entrainment weakens but continues after the ML reaches maximum depth until dissipation of the daytime boundary layer around sunset. The atmospheric entrainment characterizes the maximum of both tendencies around noon and the stronger negative moisture tendency (Barr and Betts, 1997). However, the impact of the entrainment is mainly from 7 AM to noon. Meanwhile, there is radiative cooling of the ML at all hours that there are no clouds above the ML trapping longwave radiation. The radiative cooling likely dominates afternoon when mean tendencies become negative (Fig. 4h)."**

Please rephrase and elaborate more (in red there is typo):

"Although the effect of atmospheric entrainment continues until continues until dissipation of the daytime boundary layer around sunset, it is obscured by the other contributions after noon."

➔ *The red colored typo is only in the response to reviewers document and not in the main manuscript. We apologize for the confusion..*

5. Along the results section in part 4.2 Diurnal mixing diagrams and 4.3 Climate regime dependence, hysteresis of the thermal process chain versus the moist process chain is discussed. Regarding the discussion of hysteresis, we have three comments:

   a) We highly encourage to define in this context the term hysteresis. Hysteresis is a word originally coined in science to describe systems which state depends on their history. The typical scientific example is the magnetic hysteresis. This refers to a magnet that is able to experience different magnetic moments when subject to the same magnetic field. Those magnetic moments depend on the previous states of the magnet. To us, using hysteresis in land atmospheric context may be misleading since the state of the system may be different between morning and afternoon because the external factors are also different. For instance, soil water content and vapor pressure deficit are generally different between morning and afternoon. Therefore, the sub-diurnal asymmetry may be attributed to it not because an inherent change on the interactions due to the previous history. Nonetheless, we acknowledge that hysteresis term is generally used in land-atmospheric interactions context. We recommend defining the term in this context. We already find a definition in conclusions section, line 417, the fact that "the evening path through the water-energy phase does not retrace the morning path". We would move or repeat the definition to results because there is where the hysteresis is widely discussed. In addition, we think it would be valuable to specify in which way we consider it a hysteresis. In essence, which system is subject to its previous history? Is it the vegetation, is it a vegetation-soil system? What are considered the external factors? Another simpler solution is to coin another term such as temporal asymmetry which does not imply previous history relations.

   What do you mean by a kind of hysteresis? Is it not more appropriate to call it asymmetry? Please formulate with precision

   We have changed the word "hysteresis" to "asymmetry" throughout the manuscript, and clarified what we mean by asymmetry at Lines 339-340:

   ***"There is an asymmetry in the path of moisture and temperature across the diurnal cycle, in that the extremes in the thermal process chain lead the moist process chain by 2-3 hours."***

   b) We highly recommend discussing the hysteresis' possible causes both on the land and the atmospheric coupling. We argue that due to many processes that peak at different times (e.g., radiation peaks around noon, sensible heat flux peaks in the early afternoon and latent heat flux which with peaks later in the afternoon), morning-afternoon asymmetry can be expected. It is not clear to us what is the added value of assessing the asymmetry or if the aim of the research is simply to characterize it. We recommend clarifying either if the paper aims to characterize them as a general characteristic observed or if the asymmetry is seen as a possible option to evaluate land atmosphere interactions.

It is a pity that the authors do not develop and elaborate a bit more on the physics of this asymmetry based on the observational analysis

➔ *Based on the reviewers' comment, we have tried to explain the physics of the hysteresis (asymmetry) in the land-atmosphere coupling metrics. The impact of the diurnal peaks of land heat fluxes on the leading phase of land couplings. The asymmetric behavior in the atmospheric couplings is also deeply elaborated. This description is added in Lines 348-351, 357-358, 408-409, and 411-413:*

*"H and LE peak in the early and later afternoon, respectively, each strongly controlled by gradients between the land surface and lower atmosphere. As the air warms in the afternoon and incoming solar radiation starts to decline, the thermal gradient weakens reducing H. At the same time, the warm air increases the potential evaporation by maintaining a large vapor pressure deficit, facilitating strong rates of LE."*

*"…, which reveals the phase of A(H,LCL) leading A(LE,LCL) by 2-3 hours, …"*

*"The closer to the water–limited regime, the higher the magnitude of the correlation between SWC1 and both surface fluxes."*

*"A(H,LCL) over the water–limited regime is stronger than over the energy–limited regime, which results from the larger LCL variability along with the marginal sensitivity of R(H,LCL) to the climate regime."*

Other Comments

- In line 109, the lifting condensation level is used as the variable to understand the coupling of the land with the atmosphere. We think the reader would appreciate a short sentence in which it is stated why this variable is an important indicator of the coupling to the atmosphere (e.g., because its strong relation with cloud initiation or its importance in convection schemes in atmospheric models).

  Perhaps here it is convenient to be more rigorous and stress that the condition h > LCL is a rough approximation. Majority of the situations in which shallow cumulus form are characterized by an opposite situation (h < LCL) (see for instance figure 7a at https://journals.ametsoc.org/view/journals/atsc/71/3/jas-d-13-0192.1.xml). Could they please elaborate a bit more here?

➔ *LCL is often used because it is easy to calculate from commonly available surface data, and that is also the case here. During the day when the surface inversion is broken and the mixed layer has formed, it is a reliable but rough approximation for the potential cloud base, subject to the limitations of parcel theory.*

*We thank the reviewers for pointing out the paper of van Stratum et al. (2014); we also see relevance in the Heating Condensation Framework (HCF) of A. Tawfik to explain this difference. We remind the reviewers that the general issue of the limitations of parcel theory are discussed at the end of section 3.3, but we now elaborate on the specific points regarding LCL with modified text starting at Line 122:*

*"The LCL can be characterized as a potential level of cloud base formation based on parcel theory, and is easily calculated from surface meteorological measurements, but is an approximation subject to the limitations of parcel theory. In reality, the profile of temperature and moisture above the surface also determine the level of the cloud base (Tawfik and Dirmeyer, 2014). The LCL can be compared to the planetary boundary layer (PBL) height to define an LCL*

*deficit (PBL height minus LCL; Santanello et al., 2011). When the PBL grows to the height of the LCL (corresponding to positive values of the LCL deficit), water may condense from the air parcel, and cloud formation occurs, although clouds begin to form when scattered updrafts penetrate the condensation level before the entire ML reaches the LCL (Van Stratum et al., 2014)."*

- 3.3 Mixing diagrams section. Along this section mixing diagrams are introduced. It is stated that for computing them, 2-m temperature and humidity or vapor pressure deficit are used. In the last paragraph of the section, some shortcomings of this approach are addressed. For instance, it is mentioned that embedded in this method it lies one hypothesis. The hypothesis that 2-m measurements reflect mixed-layer values. We find this hypothesis to be dubious for certain ecosystems. For instance, in vegetated areas whose trees are taller than 2-m, the measurements fall into the in-canopy range. Many forests have trees that surpasses this height. Therefore, unlike many of the observations in other land types, observations in forests lie inside the canopy. In the research 102 from 230 sites (approx., 44 %) are classified as forests. Consequently, for forests sites, we wonder how much sensitive the land and surface couplings are to the height in which the surface heat fluxes, temperature and humidity are measured. We would expect that using measurements located right above the canopy would reflect different land and atmospheric coupling. We do acknowledge the challenge of comparing the diverse land-types considered in the study within the same methodological framework. Nonetheless, we would appreciate a justification of using the 2-m height measurements for forests or at least addressing the special advantages and shortcomings of such approach for forests. In addition, we wonder how the inclusion of these observations affect the general conclusions for the land-atmospheric interactions. For instance, are patterns more easily generalizable (in figures 2, 3, 4 and 5) when forests are excluded?

In the new manuscript it has been written that the observations of FLUXNET2015 data have been assumed to be taken at 2 m above the canopy. We do not fully understand what does "assume" means here. Does it mean that it is unknown the convention of height of FLUXNET2015? For instance, is it unknown if the documentation means 2m height from the surface or 2m height above the canopy? We would like some clarification or contacting FLUXNET for further details about the height of measurements with respect to height of the trees.

→ *Canopy height and meteorological measurement height information for the flux tower observations (e.g., FLUXNET2015 and Ameriflux) is not always provided, The measurement height for such sites is assumed as the mean value across the sites that report this information. The average measurement height is 7.5 meters. The results do not change much as a result, but the component vectors in the mixing diagrams have been corrected in Fig. 4 and 5. The notation of '2-m' is replaced to 'near surface' across the manuscript and the formulation of Eq. 5 is also corrected to use the average measurement height. This description is added in Lines 133-136:*

**"As the instrument height varies among flux towers, this study computes the measurement height (h) as the difference between reported height of observation and average canopy height. When 83 observation sites (36% of the total) do not provide both heights, we assume the measurement height as the averaged value across the other available sites (7.5 m). All flux measurements are taken above the canopy while few meteorological sensors are below the canopy top."**

New comments connected to the revised manuscript

- Regarding the addition of figure 6 and section "4.4. Canopy effects", we think it adds value to the manuscript. We think figure 6(b) could be omitted because the processes explained in the section can be visualized from figure 6(a), so we do not think it adds value to the analysis. In addition, it is

a new way of visualization that has not been used before in the manuscript so it may disorientate the reader.

➔ *We agree with reviewer's comment that Fig. 6b could be omitted. Fig. 6a is now simply Fig. 6.*

- Section 5 is called "Conclusions" whereas section 6 is called "Discussion". We think it must be the other way around.

  ➔ *Based on the reviewer's comments, we changed the titles of sections 5 and 6 to "Summary" and "Conclusions", respectively.*